# Sequence-structure-function relationships in class I MHC: A local frustration perspective

Onur Serçinoğlu[1], Pemra Ozbek[2]*

1 Department of Bioengineering, Recep Tayyip Erdogan University, Faculty of Engineering, Fener, Rize, Turkey, 2 Department of Bioengineering, Marmara University, Faculty of Engineering, Goztepe, Istanbul, Turkey

* pemra.ozbek@marmara.edu.tr

**Data Availability Statement:** All allele sequences and local frustration analysis results will be available from Dryad Digital Repository after acceptance with DOI: https://doi.org/10.5061/dryad.gmsbcc2hx.

## Abstract

Class I Major Histocompatibility Complex (MHC) binds short antigenic peptides with the help of Peptide Loading Complex (PLC), and presents them to T-cell Receptors (TCRs) of cytotoxic T-cells and Killer-cell Immunglobulin-like Receptors (KIRs) of Natural Killer (NK) cells. With more than 10000 alleles, human MHC (Human Leukocyte Antigen, HLA) is the most polymorphic protein in humans. This allelic diversity provides a wide coverage of peptide sequence space, yet does not affect the three-dimensional structure of the complex. Moreover, TCRs mostly interact with HLA in a common diagonal binding mode, and KIR-HLA interaction is allele-dependent. With the aim of establishing a framework for understanding the relationships between polymorphism (sequence), structure (conserved fold) and function (protein interactions) of the human MHC, we performed here a local frustration analysis on pMHC homology models covering 1436 HLA I alleles. An analysis of local frustration profiles indicated that (1) variations in MHC fold are unlikely due to minimally-frustrated and relatively conserved residues within the HLA peptide-binding groove, (2) high frustration patches on HLA helices are either involved in or near interaction sites of MHC with the TCR, KIR, or tapasin of the PLC, and (3) peptide ligands mainly stabilize the F-pocket of HLA binding groove.

## Introduction

The sequence-structure-function paradigm plays a central role in structural biology: the primary structure (i.e. amino acid sequence) of a protein dictates the three-dimensional structure (fold), which in turn influences the function [1–3]. The close relationship between the structural architecture and function of a protein implies that disruptions to the native folded state can destabilize the protein structure, and eventually cause loss of function. In line with this perspective, several studies integrating sequence evolution with protein biophysics reported strong correlations between positional conservation levels or rates of synonymous/non-synonymous mutation and stability-related residue metrics such as solvent accessibility or hydrogen bonding patterns [4–11]. On the other hand, stability is not the main determinant of protein function. Residues that are not involved in forming the protein scaffold, such as the catalytic

**Funding:** Support from Marmara University BAP FEN-A-101018-0526 to PO is acknowledged. The funders had no role in study design, data collection and analysis, decision to publish, or preparation of the manuscript.

sites of an enzyme or binding hot spots located on protein surfaces, are important for protein function as well.

Protein evolution is thus driven by an interplay between functional and structural constraints [2,10,12,13]. With these constraints at play, mutations may occur via two alternative mechanisms: positive and negative selection [14]. In the negative (or purifying) selection, mutations leading to detrimental effects on protein stability or function are eliminated during the selection process, leading to increased levels of conservation at sites crucial for function or stability. In the positive selection model, the protein is under constant pressure to acquire "new-functions", thus change/variation in the amino acid sequence is favored, especially at sites with direct relevance to protein function [15].

From this perspective of molecular evolution and protein function, the human class I Major Histocompatibility Complex (MHC I, also named the Human Leukocyte Antigen, HLA) is an interesting system [16–19]. MHC I consists of heavy (H) and light (L) chains, with the L chain being an invariant protein partner called β-2 microglobulin (β2m), respectively (Fig 1). Following assembly within the Endoplasmic Reticulum (ER), the HLA is unstable in the absence of a peptide ligand [20,21]. Intracellularly derived antigenic peptides originating from self or foreign proteins are loaded into the binding groove of HLA with the help of a multiprotein complex called the Peptide Loading Complex (PLC). Here, the peptide contacts six different binding pockets (A,B,C,D,E and F) within the peptide binding groove of HLA [22].

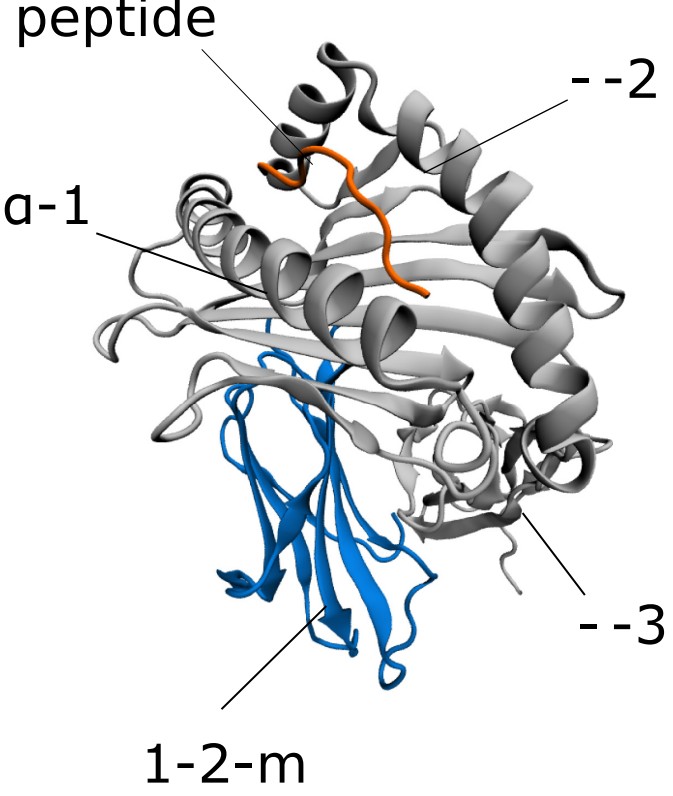

**Fig 1. The three-dimensional structure of the pMHC.**

Upon the formation of a stable peptide-loaded MHC (pMHC), the complex is transported to the cell surface, and presents peptide ligands to immune cell receptors such as the T-Cell Receptor (TCR) of cytotoxic CD8+ T-cells, and Killer-cell Immunoglobulin-like Receptors (KIRs) of Natural Killer (NK) cells [23–26].

## The structure of HLA-B*53:01 complexed with peptide TPYDINQML (PDB code: 1A1M) is shown for demonstration purposes

With more than 10000 identified alleles, HLA I is the most polymorphic protein in humans [27]. The HLA I protein is clearly undergoing a process of positive selection, where the acquisition of new variants modulate the function of antigen presentation via modifying the peptide ligand recognition specificities [16,28–31]. In other words, functional constraints require HLA to maintain a very high level of allelic diversity. This allelic diversity helps the immune system cover a large space of potential peptide binders, and thereby fight against pathogens effectively [16,19,24,32]. On the other hand, HLA genes are also strongly linked to infectious and autoimmune diseases due to high polymorphism levels [33].

Despite this high sequence variation in HLA and the enormous diversity of peptide ligands, experimentally elucidated pMHC structures of different alleles display an "ultra-conserved" fold with minimal variation [34]. Moreover, most of the TCR-HLAcrystal structures display a conventional docking mode of TCR over the HLA [35,36]. On the other hand, HLA allelic diversity may affect interactions with other proteins as well. Only certain allele groups containing specific epitopes interact with KIR [25,37]. Structural details of the PLC-pMHC interaction were also only recently revealed [34,38–41].

HLA polymorphism should be taken into account in order to truly comprehend the mechanisms involved in the formation of stable human pMHC and interactions with TCR and KIR molecules. However, incorporating such extreme levels of polymorphism into wet-lab experiments is unfeasible. Hence, computational biophysics methods are preferred in large-scale modeling studies. In particular, homology modeling was used to classify a high number of HLA alleles into distinct groups based on peptide interaction patterns [42], binding pocket similarities [43] and surface electrostatics [44]. These studies mainly focused on classifying HLA alleles based on their peptide-binding or protein interaction behaviors, and therefore only the peptide-binding groove was modeled. However, characterization of the relationship between HLA polymorphism and complex stability as well as TCR/KIR/PLC interactions requires the modeling of the whole complex and the use of proper methods for identifying residue-level effects on stability and protein interactions. To this end, quantification of local frustration in the structure can be utilized [45–49].

The concept of frustration within protein structures is useful to establish links between their biological behavior and structures. The energy landscape theory indicates that a given chain of amino acids fold into a native structure through an ensemble of structures, and the resulting native state is a minimally frustrated heteropolymer [50]. In other words, interacting energies between the building blocks of the protein structure are minimized within the native state, leading to a well-defined three-dimensional shape. Yet, even in the native state, frustration may exist locally at multiple sites within the structure. Computationally, the local frustration analysis is based on calculation of pairwise contacts between amino acids using an appropriate force-field, and comparison of these contacts to possible alternative contacts made by other amino acid pairs at each site in the structure. Briefly, the analysis quantifies the degree of energetic optimization of the contacts made by each residue, and thus how much local frustration is present at a given site. Given the extreme level of polymorphism in HLA, such information may be particularly useful for establishing a link between functional roles of each

residue (e.g. whether they are important for folding or binding), and the respective level of polymorphism observed within a sequence-structure-function context. Local frustration analysis has already been applied successfully to perform such sequence-structure-function studies on calmodulin [51,52], repeat proteins [5,53], and TEM beta-lactamases [6].

Here, we provide an analysis of local frustration patterns of 1436 HLA I alleles. We explain how class I MHC retains a conserved fold while maintaining highly versatile ligand specificities using homology-modeled human pMHC structures. Using the frustration data, we also show the existence of local frustration-based energetic footprints of polymorphism, providing a biophysical basis for the previously observed differences between molecular stabilities/cell surface expression levels of different allele groups and TCR/KIR/PLC interactions. Finally, we also provide a local frustration based explanation of how peptides stabilize peptide binding pockets.

## Results and discussion

### Sequence variation in HLA binding groove and α-3 domain

We began with an analysis of sequence variation in the HLA I peptide binding groove. A total of 8696 HLA I binding groove (α-1 and α-2 domains) sequences (including 2799, 3433 and 2464 alleles from HLA-A, -B, and -C gene loci, respectively) were included in the analysis. We identified amino acid variation at each respective position in the HLA peptide binding groove by constructing sequence logos (Fig 2A). In line with findings of a recent analysis [54], we detected high levels of variation at most binding groove positions. However, some positions were relatively conserved, and dominated by a single amino acid. This dominance is due to the imbalance between the number of occurrences of the respective amino acid and those of the others. Glycine, phenylalanine, proline, tryptophan, leucine, aspartic acid and arginine were found to be the most conserved amino acid types.

The sequence logos also display several "hyper-variable" positions as well (positions 9, 24, 45, 67, 97, 116, 138, 152, 156, and 163). This variation is expected, since all of these positions are located within the peptide binding pockets, and were previously shown to define allele-specific peptide ligand repertoires [42,43,55].

We next quantified the evolutionary importance of each position in the binding groove by computing the real-value Evolutionary Trace (rvET) ranks per position [56]. rvET is an absolute rank of a given position in terms of its evolutionary importance. Here, lower rvET ranks/scores indicate higher evolutionary importance and vice versa. The ET analysis here is based on both sequence variation/conservation and the closeness of sequence divergence to the root of the constructed phylogenetic tree at a given sequence position. Thus, while conserved positions in a multiple sequence alignment tend to have low rvET scores, relatively less conserved positions may also obtain low rvET scores as well, provided that the sequence divergence occurs near the root of the tree and variation occurs within small rather than large branches of the tree [56–59]. As such, this method is particularly suitable for analysis of variation in HLA sequences from different gene loci which may feature. In general, we observed higher rvET values at positions with the highest sequence variation and vice versa (Fig 2B).

We repeated the above sequence variation and evolutionary trace analysis for the α-3 domain of HLA as well, and generated sequence logos and computed rvET scores. Here, the number of allele sequences used as input was lower (1436) since the α-3 domain sequences for many alleles are unavailable (see section below as well). As expected, the level of polymorphism of the α-3 domain is lower than that of the binding groove, as demonstrated by the dominance of hydrophobic amino acids in many positions and lower rvET scores (Fig 3).

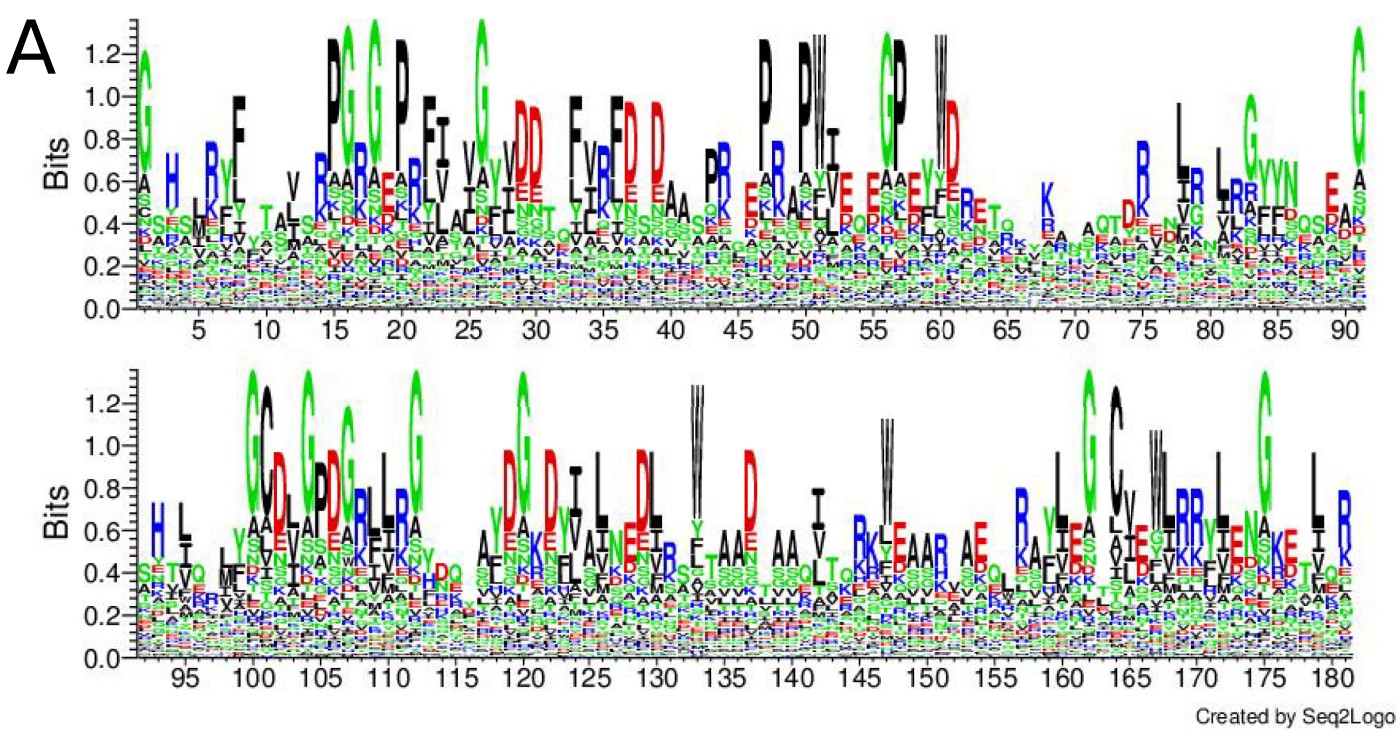

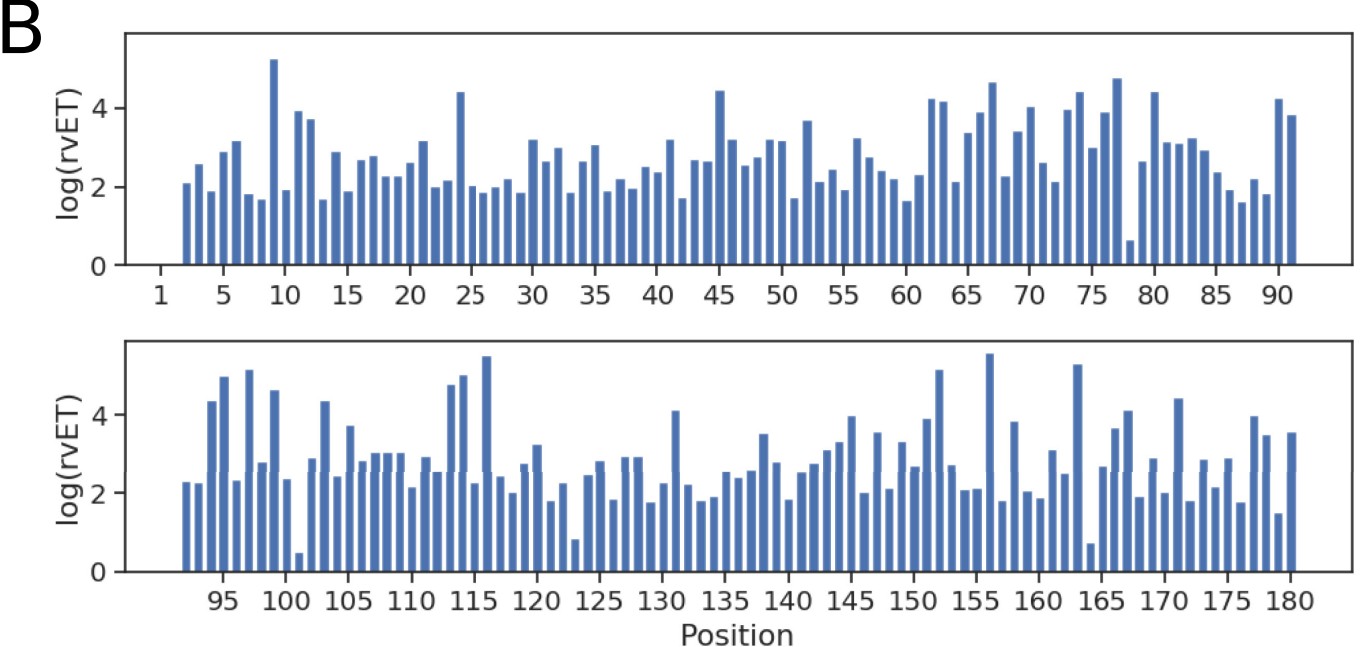

**Fig 2. Sequence conservation/variation and evolutionary importance of HLA I peptide-binding groove positions.** (A) Sequence logo of the HLA I peptide-binding groove (residues 1–180). Polar, neutral, basic, acidic and hydrophobic amino acids are colored green, purple, blue, red, and black, respectively. (B) real-value Evolutionary Trace (rvET) scores of binding groove positions. Low rvET scores indicate high evolutionary importance, and vice versa.

# A

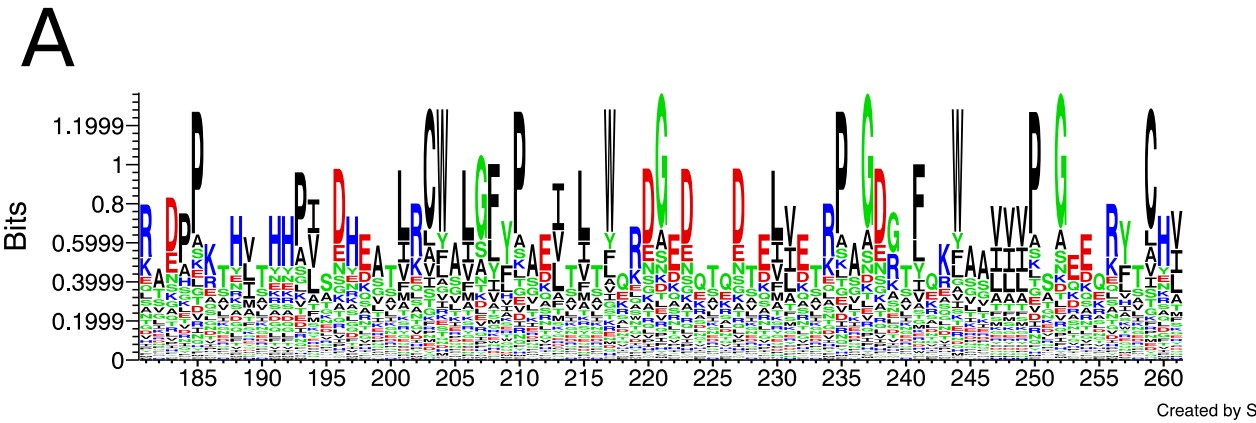

Created by S

# B

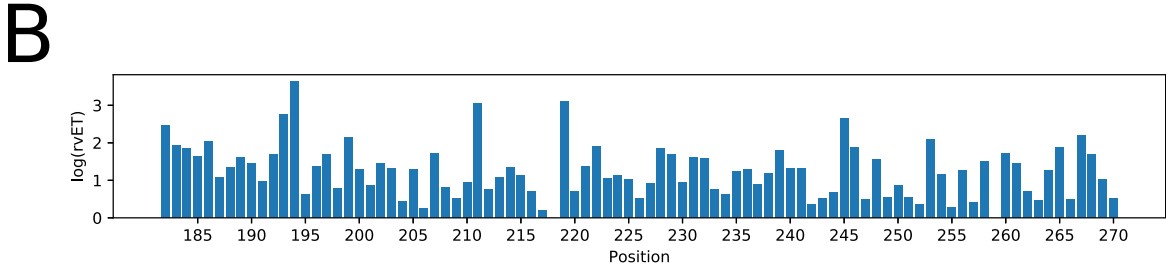

**Fig 3. Sequence conservation/variation and evolutionary importance of HLA I α-3 domain positions.** (A) Sequence logo of the HLA I α-3 domain (residues 181–261). Polar, neutral, basic, acidic and hydrophobic amino acids are colored green, purple, blue, red, and black, respectively. (B) real-value Evolutionary Trace (rvET) scores of α-3 domain positions. Low rvET scores indicate high evolutionary importance, and vice versa.

## Homology modeling of HLA alleles in the context of pMHC

Next, we investigated how HLA sequence variation is reflected on the complex structure. As reported previously [34,43,44,60], the number of experimentally determined HLA structures with different alleles is significantly lower than the total allele number. Hence, a homology modeling approach is necessary. Moreover, homology models of the full complex, including the binding groove as well as distant β2m and α-3 domains, are needed, since these domains make extensive contacts with each other as well as the rest of the structure, and were previously shown to be essential for peptide binding [61,62]. However, the sequences of the α-3 domain of a majority of HLA alleles have not been identified yet. It is also necessary to model peptides within the binding groove as peptide ligand is an integral part of the HLA structure. Similar to limitations in HLA structures, the number of identified peptide ligands with binding affinity measurements for individual HLA alleles is limited as well, with some alleles having no peptide ligands identified so far [63]. Therefore, it is also necessary to predict peptide ligands using computational approaches. We thus selected 1436 HLA alleles (464, 689 and 283 from HLA-A, -B and -C loci, respectively) with complete HLA sequence (including all three domains α-1, α-2, and α-3) for homology modeling (the complete list is given in S1 Table). The list of homology modelled alleles included 41 out of 42 core alleles representing the functionally significant sequence variation [54]. We predicted up to 10 strong binder peptides for each of the 1436 alleles using netMHCpan 3.0 [64], and obtained homology models in the context of these peptides. Note that it is also possible to perform more complicated HLA-specific peptide docking or modeling [65,66] to generate more reliable binding modes for peptide ligands. Since our

main focus was a characterization of the effect of polymorphism, we nevertheless used homology modeling to model the peptide ligands within the complex for practical reasons.

## Integrating biophysics into HLA I evolution

After generating the homology models, we analyzed the local frustration within pMHC structures. A Single Residue Frustration Index (SRFI) was obtained as a position-specific local frustration score: amino acids with optimized energetic contacts are minimally frustrated, where those that are the least preferred at their respective positions are highly frustrated. If neither, then the frustration is termed "neutral". SRFI values thus indicate how ideal (minimally frustrated) the contacts of each position are or how much frustration is present (highly frustrated).

We used the *frustratometer2* tool [67] to quantify SRFI at each position in homology models. Since multiple peptides (and hence structures) were modeled for each allele, we calculated median SRFI per allele per position, and used these median SRFI values in further analyses.

In order to get an overview of local frustration against evolutionary importance of each position, we first mapped position-specific median SRFI and rvET values onto the HLA I binding groove (Fig 4). SRFI distributions of several selected positions are also given in Fig 5A. As expected, the top two minimally frustrated positions were cysteines responsible for forming the conserved disulfide bridge between positions 101 and 164. Mutations at these positions abolish HLA expression [27,68]. Minimal frustration was also observed at conserved positions

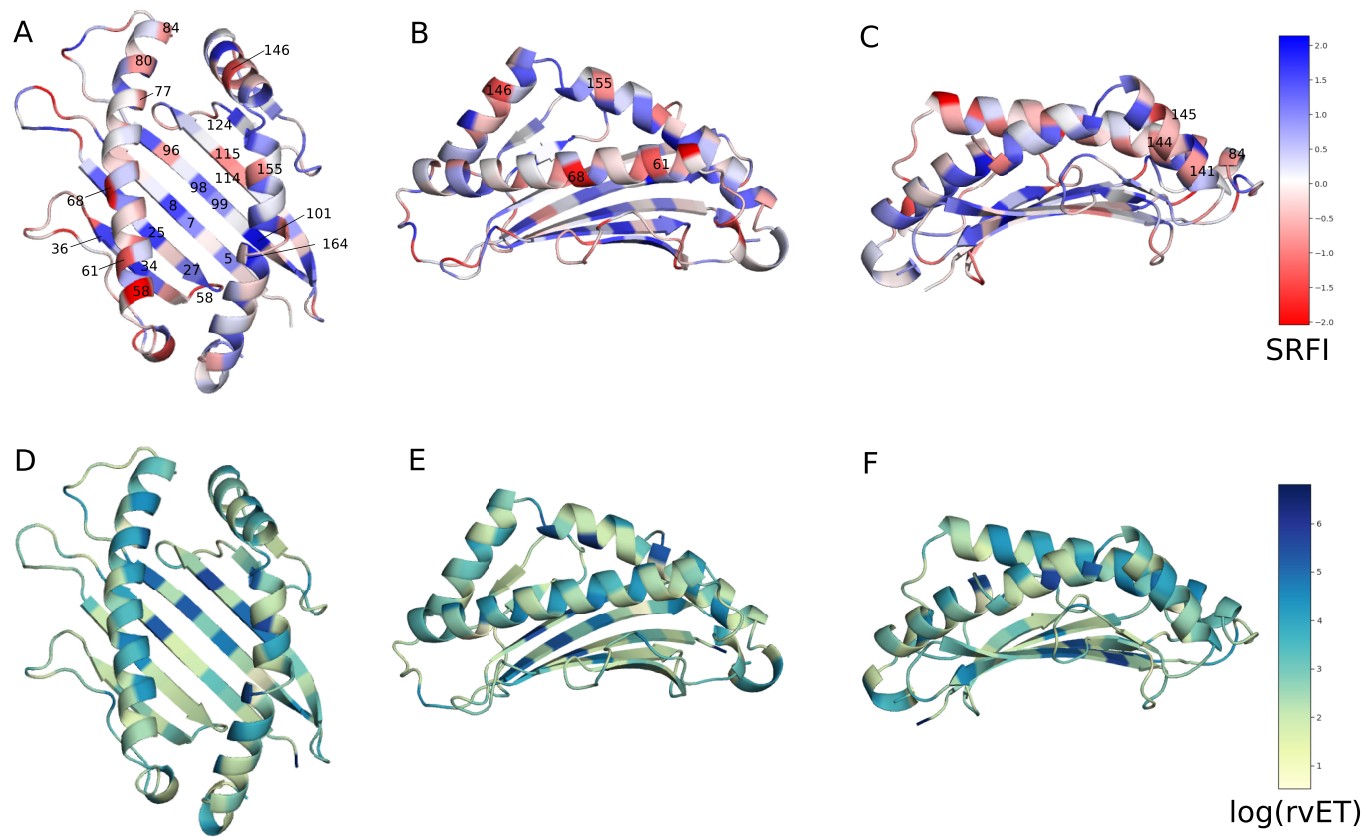

**Fig 4. Position-specific Single Residue Frustration Index (SRFI) and rvET scores mapped onto the HLA I binding groove.** (A, B, C) SRFI mapped onto binding groove positions. (D, E, F) log(rvET) mapped onto binding groove positions. Selected positions are also shown on the structure.

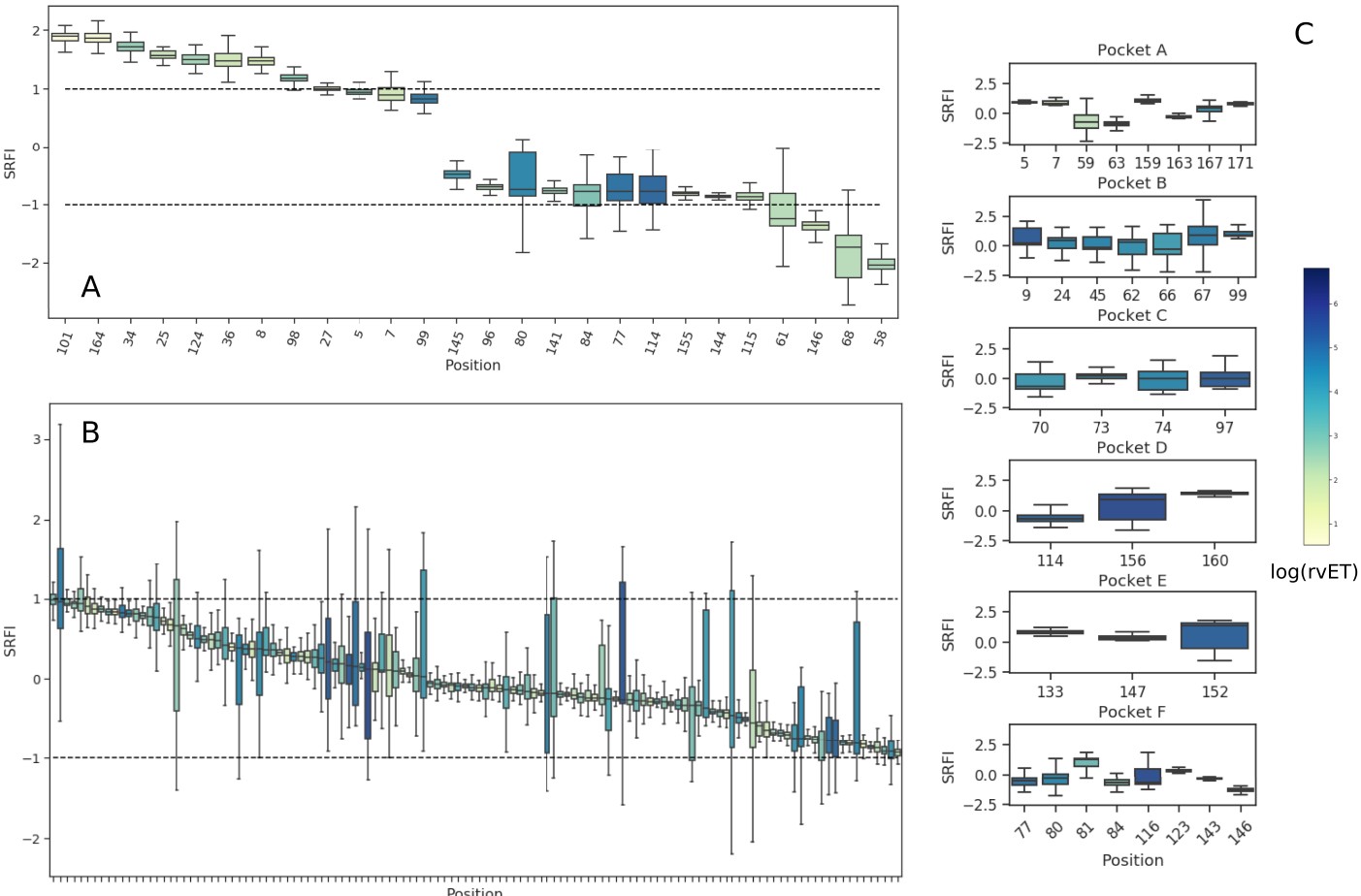

**Fig 5. Box-plots of position-specific SRFI values for the HLA I peptide-binding groove.** (A) SRFI box-plots of selected minimally and highly frustrated residues within the HLA peptide binding groove (B) SRFI box-plots of positions with neutral frustration. (C) SRFI box-plots of peptide-binding pocket positions. Coloring according to log(rvET) values.

located in the β-sheet floor of the binding groove or in α-1 and α-2 domains contacting the β-sheet floor (5, 7, 8, 25, 27, 34, 36, 101, and 164) (Fig 4A). Our observation of minimal frustration at conserved positions are in line with recent findings of Dib et al. [69], where co-evolving residues were shown to avoid residues important for protein stability. Haliloglu et al. previously analyzed several HLA structures using the Gaussian Network Model (GNM), and identified positions 6, 27, 101, 103, 113, 124 and 164 as possibly important for stability [70]. Here, we observed minimal frustration at either these or their sequence neighbors.

Next, we identified minimally- and highly-frustrated positions within the α-3 domain by mapping position-specific median SRFI as well as rvET values onto the α-3 domain structure and plotting SRFI distributions (Fig 6 and Fig 7). Median SRFI and rvET values can be found in S2 Table as well. Here, we observed the same relationship between local frustration and sequence conservation: several positions on α-3 beta-sheets were found to be minimally-frustrated and conserved, including positions 204, 203, 247, 259, 261, 215, and 257.

Our observation of minimal frustration at conserved positions within the peptide binding groove and α-3 domain indicates that the interactions made by the residues in these positions of homology models are energetically favorable (thus minimal frustration), and therefore

responsible for the overall structural stability. In other words, the importance of these residues for structural stability may explain their high conservation. Fig 8 shows how the residues at these positions are structurally connected to each other within the HLA-B*53:01 structure

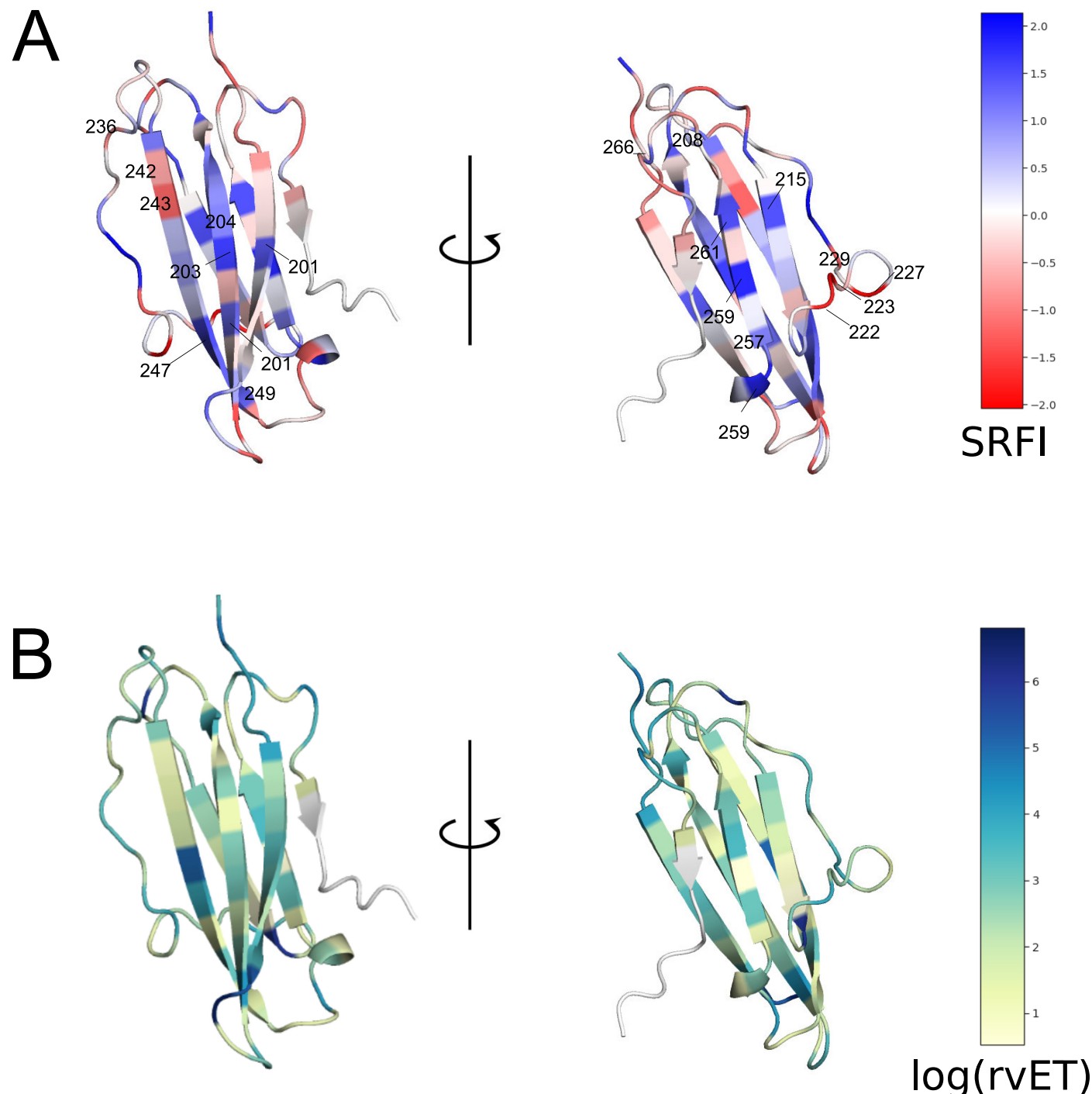

**Fig 6. Position-specific Single Residue Frustration Index (SRFI) and rvET scores mapped onto the HLA I α-3 domain.** (A) SRFI mapped onto α-3 domain positions. (B) log(rvET) mapped onto α-3 domain positions. Selected positions are also shown on the structure.

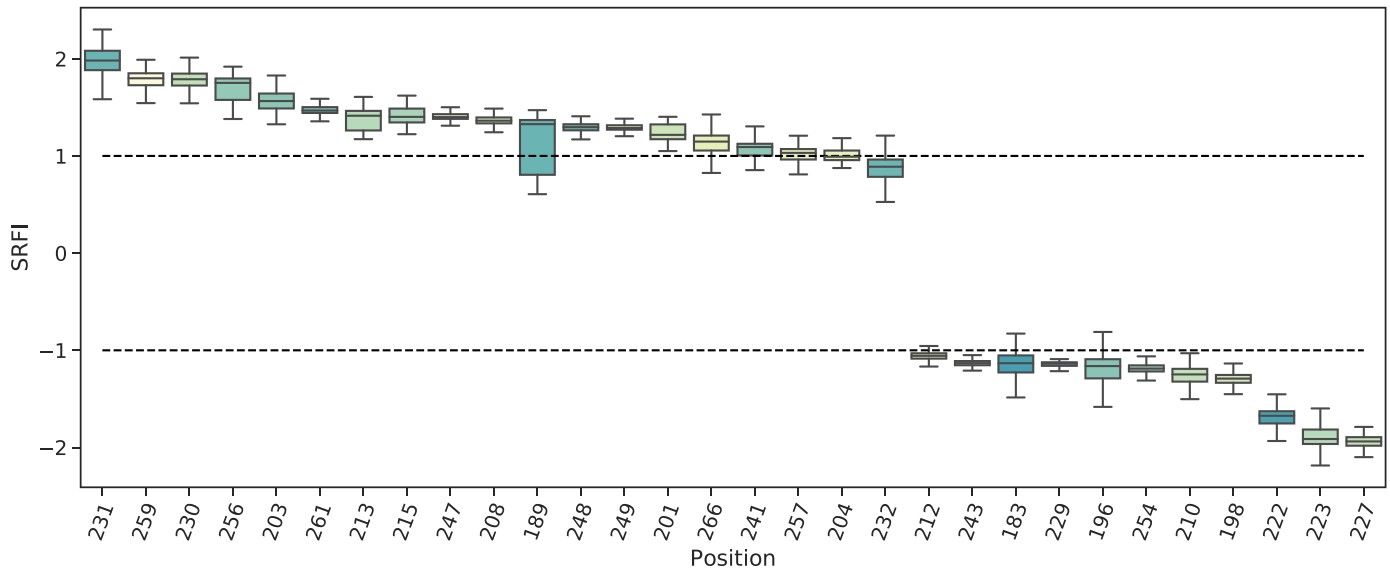

**Fig 7. Box-plots of position-specific SRFI values for the HLA I α-3 domain.** SRFI box-plots of minimally- and highly-frustrated residues within the HLA α-3 domain. Coloring according to log(rvET) values.

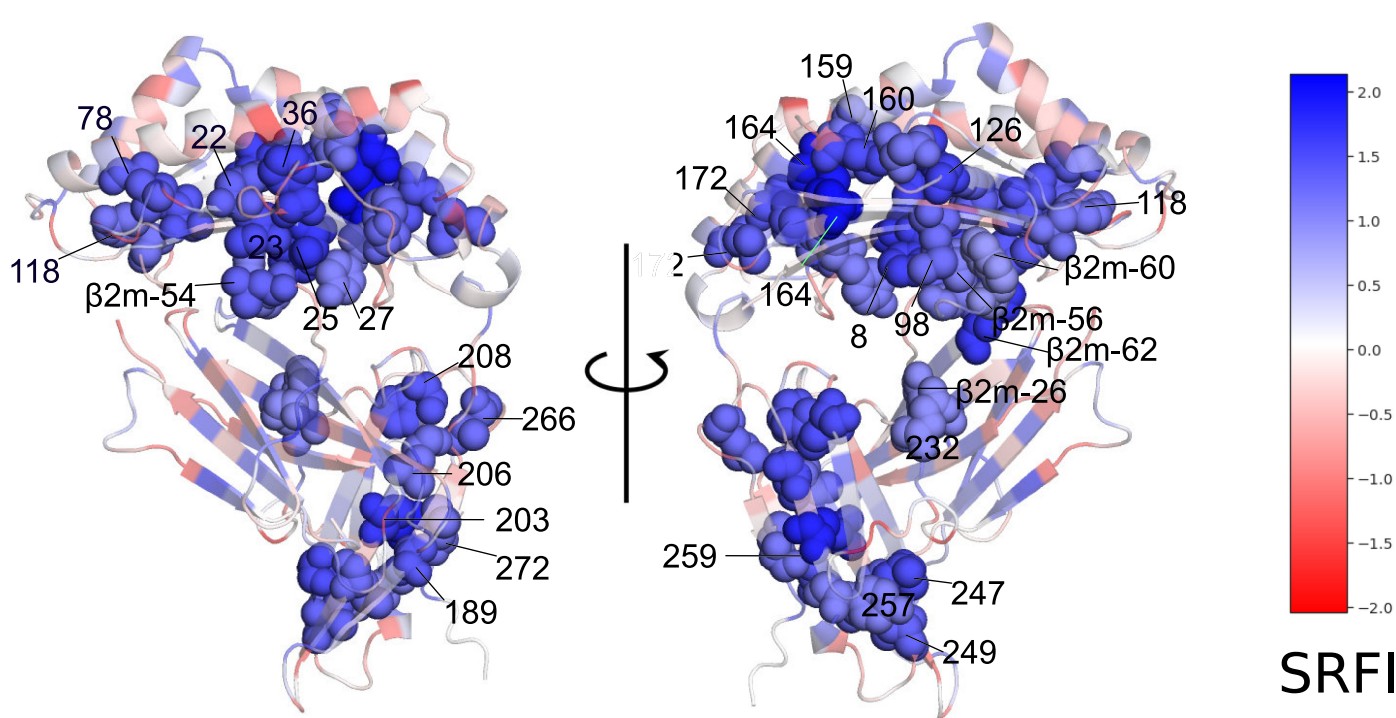

**Fig 8. Minimally-frustrated and conserved positions within the HLA structure.** Residues are drawn in van der Waals spheres representation. Coloring according to SRFI value. The structure of HLA-B*53:01 (1A1M) was used for demonstration purposes. Selected β2m residues are also shown in spheres to highlight interaction with HLA.

(pdb code 1A1M, used for demonstration purposes) (Fig 8). Here, most of the residues can be observed to form physical clusters within individual domains. Minimal frustration at binding groove residues contacting the β2m (8, 25, 27, and 98) as well as β2m residues 56, 60 and 62 clearly points to conserved interactions at the HLA-β2m interface that are likely important for structural stability. On the other hand, no interacting pair of minimally frustrated residues were detected at the β2m and α-3 domain interaction interface. Here, the lowest frustration level was that between residues 232 and β2m-26.

We also observed several positions on α-1 and α-2 helices and on the α-3 domain loops with relatively high frustration levels (58, 61, 68, 80, 84, 141, 144, 145, 146, 155, 222, 223, and 227) (Fig 4, Fig 7). High frustration at a position may indicate that the interactions made by the respective amino acid is energetically destabilizing compared to all other amino acids. These highly frustrated residues may thus be involved in protein or ligand binding. Indeed, most of these positions are located within interfaces of interaction with either TCR or tapasin of the PLC.

The interaction between the TCR and pMHC I is a central event in adaptive immunity [26,71,72]. The TCR recognizes a peptide antigen only when presented by an MHC molecule (MHC restriction) [72,73]. TCR-pMHC structures determined to date indicate a conserved diagonal binding geometry, where the hypervariable CDR3 loop of TCR contacts the peptide, and germline-encoded variable α and β domains (Vα and Vβ) contact HLA I α-1 and α-2 helices, respectively [74]. Although deviations from this conventional docking mode exist, including a reversed polarity docking mode [75,76], TCR signaling in such unconventional modes is limited [77]. The importance of the conventional docking mode for TCR signaling adds support to the germline-encoded model of TCR-pMHC interaction, in which evolutionarily conserved TCR-pMHC contacts were proposed to govern MHC restriction [78]. In this regard, previous analyses of TCR and human pMHC structures highlighted the importance of contacts made by HLA α-1 and α-2 helix residues 69 and 158 [79] or 65, 69 and 155 as a "restriction triad" [80]. We observed high SRFI levels at or near these positions.

Local frustration data can also provide a basis for PLC-MHC interactions. Tapasin is the main component of the PLC which contacts the HLA binding groove. Predicted tapasin-MHC interactions [39,81] and a recently elucidated crystal structure of pMHC with TAPBPR (a tapasin homolog) [82–84] indicated that tapasin cradles the MHC molecule via contacts with α-3 domain and $\alpha_{2-1}$ helix segment. Our observation of high SRFI levels at positions 141, 144, 145, and 146 located on the $\alpha_{2-1}$ helix segment as well as at 222, 223, and 227 on the α-3 domain is in line with these findings.

Median SRFI values of most binding groove positions were found to indicate neutral frustration (Fig 5B). On the other hand, HLA polymorphism at peptide binding pocket positions apparently caused significant drifts towards minimal or high frustration as well, even though the median SRFI remained within the neutral frustration range (-1 to 1).

Overall, these results suggest that the human MHC evolves to maintain its structural fold, as evidenced by the dominance of conserved core positions showing minimal frustration within the HLA peptide binding groove. Moreover, the presence of relatively higher frustration at or near TCR and tapasin contact positions on α-1 and α-2 helices may also provide a biophysical basis for protein interactions of pMHC.

## Clustering of HLA alleles into distinct groups based on frustration data

Next, we performed a hierarchical clustering analysis of allele-specific SRFI profiles. For simplification, we excluded positions with less than 0.5 SRFI variation from the analysis. Clustering results are shown in Fig 9A, and complete lists of alleles in each cluster are given in S1

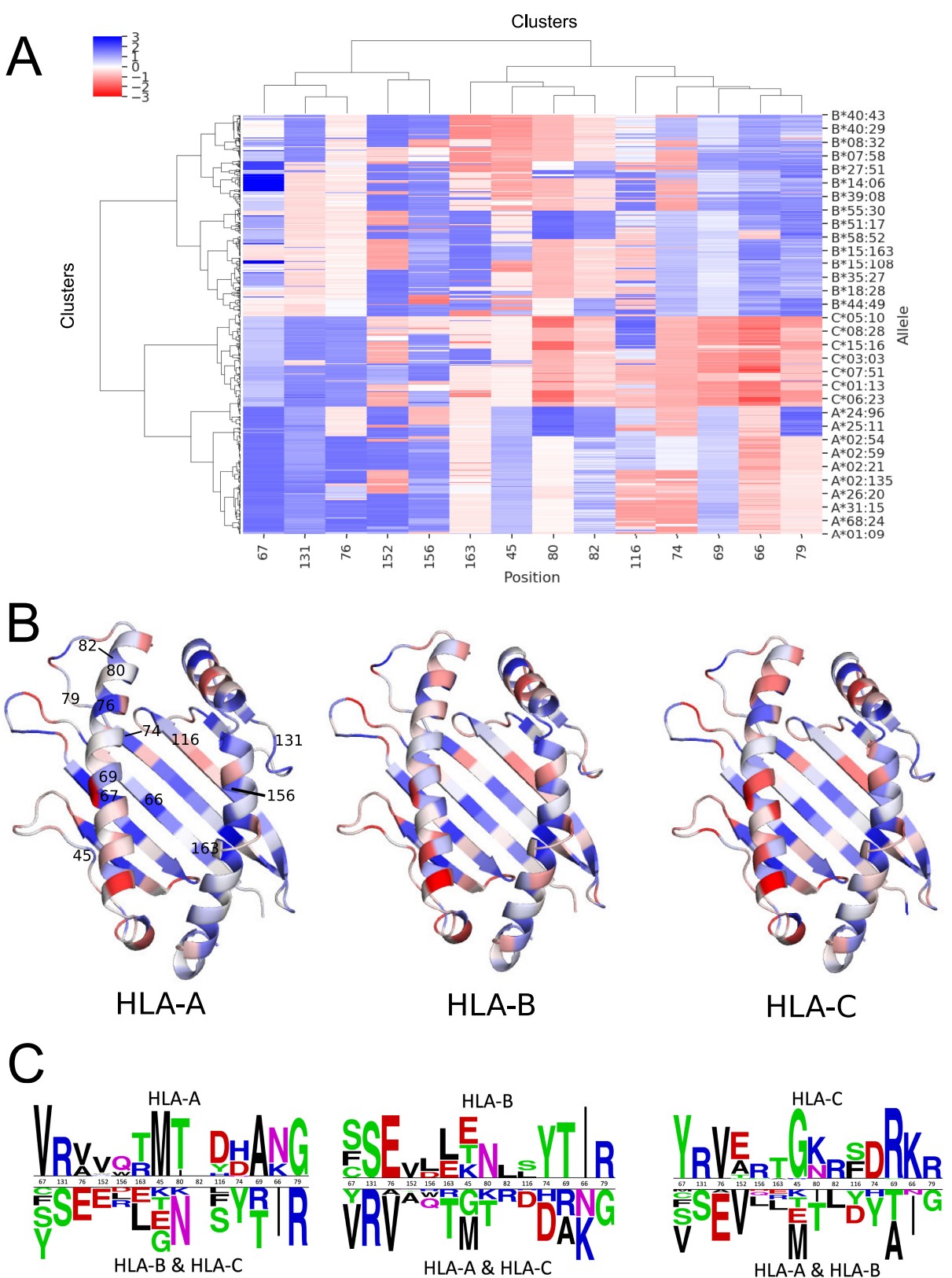

**Fig 9. Clustering of allele-specific SRFI profiles based on 14 binding groove positions.** (A) SRFI cluster heatmap. Clustering was performed for all 1436 alleles included, yet not all of these alleles are indicated on axis label for clarity. (B) SRFI of three identified clusters corresponding to three HLA gene loci mapped onto the binding groove. Coloring as in Fig 5. (C) Sequence logos of clustering positions to highlight amino acid differences between allele in different clusters. The logos were generated using Two Sample Logo server [86].

Table. Strikingly, alleles from different gene loci were clustered into three separate groups based on frustration data of only 14 positions (45, 66, 67, 69, 74, 76, 79, 80, 82, 116, 131, 152, 156, 163). An exception was the HLA-B*46 group, which was clustered along with the HLA-C alleles (S1 Table. This is not surprising, as alleles in this group share the KYRV motif at 66, 67, 69, and 76 with HLA-C [85].

Clustering of HLA alleles from distinct loci into separate groups has been previously demonstrated based on peptide binding pocket similarities [43], peptide-HLA contacts [42], surface electrostatics [44] and peptide binding repertoires [87]. Here, HLA-A, -B and–C alleles were clustered into distinct groups from a different perspective using local frustration data. Moreover, the structural energetics aspect provides an additional level of detail.

Unlike data used in previous studies, the SRFI may explain previously reported differences between HLA alleles in terms of complex stability and hence, the cell surface expression levels. Compared to HLA-A and HLA-B, lower cell surface expression levels were previously observed for HLA-C alleles [88–91]. Moreover, peptide repertoires of HLA-C alleles are known to be more limited [88,92]. KYRV motif was also shown to be responsible for intrinsic instability of HLA-C [90]. Relatively higher frustration in HLA-C group, especially at binding pocket positions of 66, 74 and 80, is in line with these findings: higher frustration may introduce a destabilizing effect into the binding pockets, lead to restrictions in peptide binding, and thereby reduce cell surface expression levels. Among these positions, 66 is a member of the B pocket and increased frustration in this position may indicate a less stable B pocket in HLA-C. Likewise, position 80 is a member of F pocket and a similar effect may be valid for the F pocket as well. Around position 80, positions 79 and 82 also display high frustration as well, further contributing to F pocket frustration.

HLA-C additionally differs from HLA-B and HLA-A in terms of interactions with KIRs of NK cells [93,94]. The alleles in this group contain either the C1 or C2 epitopes with an arginine or lysine at position 80, respectively [95]. Moreover, HLA-C (and HLA-B46, which is clustered alongside HLA-C) is distinguished by a KYRV motif at positions 66, 67, 69 and 76 [92]. Previous crystal structures of HLA-C and KIR clearly show that the KIR contacts residues on α-1 helix of HLA near peptide C terminus [96,97]. High levels of frustration in this contact area of HLA-C (Fig 9B) may explain why HLA-C better interacts with the KIR than HLA-A and HLA-B alleles.

All in all, these results support the view that HLA-C is intrinsically less stable on the cell-surface via a post-translational mechanism [88,90]. This mechanism may simply involve a less than optimal packing in the binding groove of HLA-C. Our results may also explain the limited diversity of HLA-C peptide ligands: higher frustration in ligand binding sites may indicate a lower peptide binding capability.

## Effect of peptide binding on local frustration profiles

Peptide-free (empty) MHC does not have a well-defined 3D structure (no crystal structure of a peptide-free MHC could be obtained so far). Instead, empty MHC continuously switches between alternative conformations until a sufficiently stable peptide-HLA interaction is achieved [21]. Peptide binding to HLA occurs via six pockets in the binding groove named A, B, C, D, E and F [22]. For many alleles, A, B and F pockets are decisive for peptide binding (A/

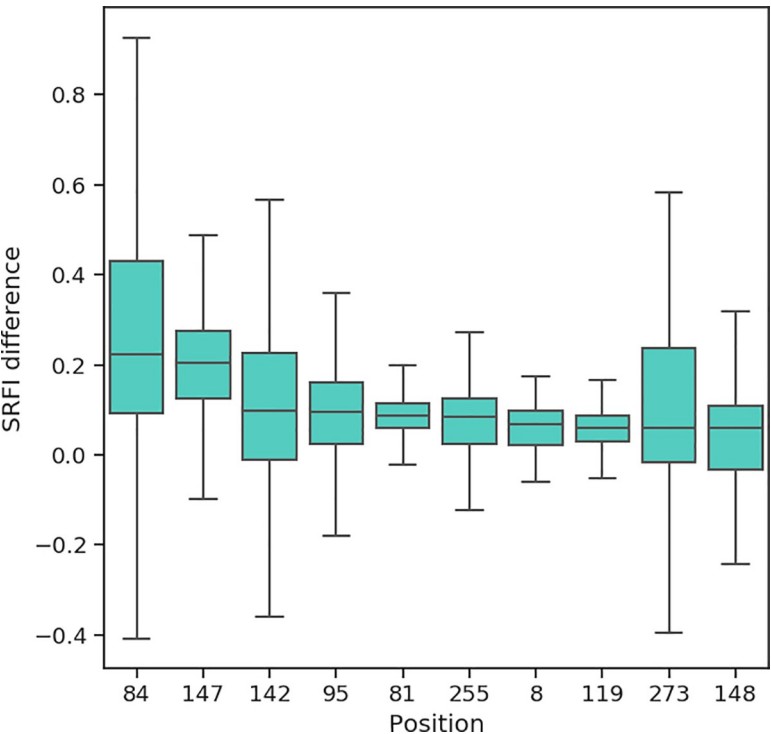

**Fig 10. Box-plots of SRFI change upon peptide binding.** Positions are ordered from left to right according to decreasing change in median SRFI values. Note that the plot only includes top 10 positions showing the highest increase in SRFI levels upon peptide binding.

B and F pockets binding N- and C-terminus of peptide ligands, respectively). Structural integrity of the F pocket of some alleles was previously shown to be more sensitive to peptide truncation in MD simulations than those of A and B pockets [41,98–100]. This implies that contacts between peptide C-terminus and HLA F pocket are highly important for molecular stability. We reasoned that, by comparing peptide-free and -loaded MHC frustration profiles, the positions that depend on peptide contacts least/most for stability can be identified. Thus, we additionally quantified local frustration in peptide-free homology models, and calculated changes in SRFI upon binding of the peptide ligands to each allele. For each residue, we computed the difference between the median SRFI from peptide-loaded structures and the SRFI value in a single peptide-free structure of each allele. A single SRFI difference value was then obtained for each allele, and the distributions of these differences are shown in Fig 10. In line with the findings of previous studies, SRFI increase (hence reduction in frustration) upon peptide binding was highest near the F-pocket (positions 81, 84, 95, 142, and 147). Nevertheless, we should emphasize that our approach here involved a straightforward comparison with simple removal of peptides from the binding groove to generate peptide-free HLA. The number of peptides for each allele was also very limited. Further studies, possibly involving conformational differences between peptide-loaded and -free forms, are necessary for a more proper investigation of frustration changes upon peptide-binding in HLA.

## Local frustration in the context of TAPBPR-HLA interaction

We also investigated the existence of a possible relationship between local frustration data and dependence of different HLA alleles on cofactors for structural assembly of the complex. For

this purpose, we used the results of a recent study by Ilca et al. [101] as reference, where the authors reported a stronger preference of the TAPBPR for HLA-A molecules (especially those that belong to A2 and A24 HLA supertypes [87]) than for the HLA-B and HLA-C molecules. Here, local frustration may be relevant with respect two aspects of TAPBPR-HLA interaction. First, the local frustration profiles of peptide-binding pocket positions may describe relatively higher or lower stability of peptide binding grooves, and hence influence TAPBPR dependency. Second, the interface of HLA with the TAPBPR molecule may exhibit differential frustration between TAPBPR binders and non-binder alleles. Similar to the clustering analysis described above, we performed SRFI-based clustering of either peptide-binding pocket or TAPBPR interface residues of 30 top TAPBPR binder HLA allotypes reported by Ilca et al. The cluster maps are shown in Fig 11.

We found that the top 8 TAPBPR binder HLA alleles, A*68:02, A*02:03, A*02:01, A*02:06, A*23:01, A*24:02 and A*24:03 were grouped together. They were further clustered into separate groups including (1) A*68:02, A*02:03, A*02:01, A*02:06 and (2) A*23:01, A*24:02, A*24:03. Ilca et al. highlighted the presence of H114/Y116 pair within the F pocket as a prerequisite for strong TAPBPR interaction. Fig 11A indeed shows that the frustration level of 116 in TAPBPR binders is indeed distinct from those in non TAPBPR binders. On the other hand, non TAPBPR binders also featured different SRFI levels in this position, ranging from minimal frustration in several HLA-C and HLA-B alleles to high frustration in many other HLA-A alleles which featured D116. As such, there was no correlation between the SRFI level of position 116 and TAPBPR binding preference. There were several other positions within the binding pockets that exhibited distinct frustration levels in TAPBPR binders, including 97, 80, and 84. Y84 within the TAPBPR-MHC I complex was found to interact with the E102 of TAPBPR, instead of K146 of α-2-1 helix region and C-terminus of the bound peptide as in the unbound state [102]. Increased frustration at this position in TAPBPR binders, except for A*23:01, A*24:02, and A*24:03, may contribute to their higher binding tendency (Fig 11A and 11B). Interestingly, the increase in frustration was not due to an amino acid difference, as both TAPBPR binders and non-binders included Y in this position. This may be due to an allosteric effect from the bottom of the F pocket: the presence of Y at 116 may influence peptide-HLA contact patterns, which may cause a slight change in local frustration of Y84. Nevertheless, the effect on Y84 may be insufficient to not explain differences in TAPBPR association, as three TAPBPR binders (A*23:01, A*24:02, and A*24:03) exhibited lower frustration than other binders. On the other hand, all TAPBPR binders showed lower frustration for residue 128 within the binding interface (Fig 11B). Such perturbations may influence the bond-making potentials of interface residues, and may provide possible directions to investigate chaperone dependence using computational biophysics methods in future studies.

## Material and methods

### Sequence variation analysis

Aligned amino acid sequences of the α-1 and α-1 domains (i.e. the binding groove) of the HLA were retrieved from the IMGT/HLA database [27]. The sequences included in this file were converted to FASTA format for further analysis using Biopython [103]. Seq2Logo 2.0 web server was used to generate sequence logos to indicate conservation/variation at specific sites [104] using Shannon's Information Content (IC) [105] as follows:

$$I = \sum_a p_a \cdot log_2 p_a / q_a$$

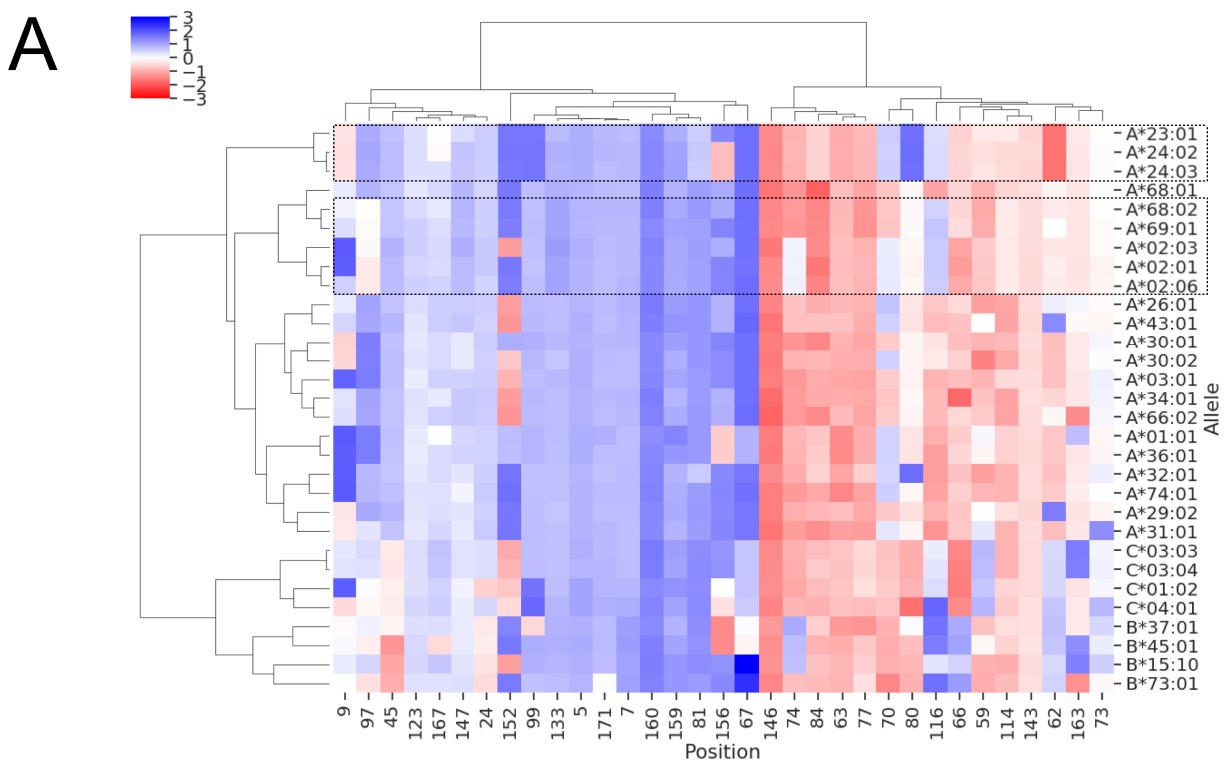

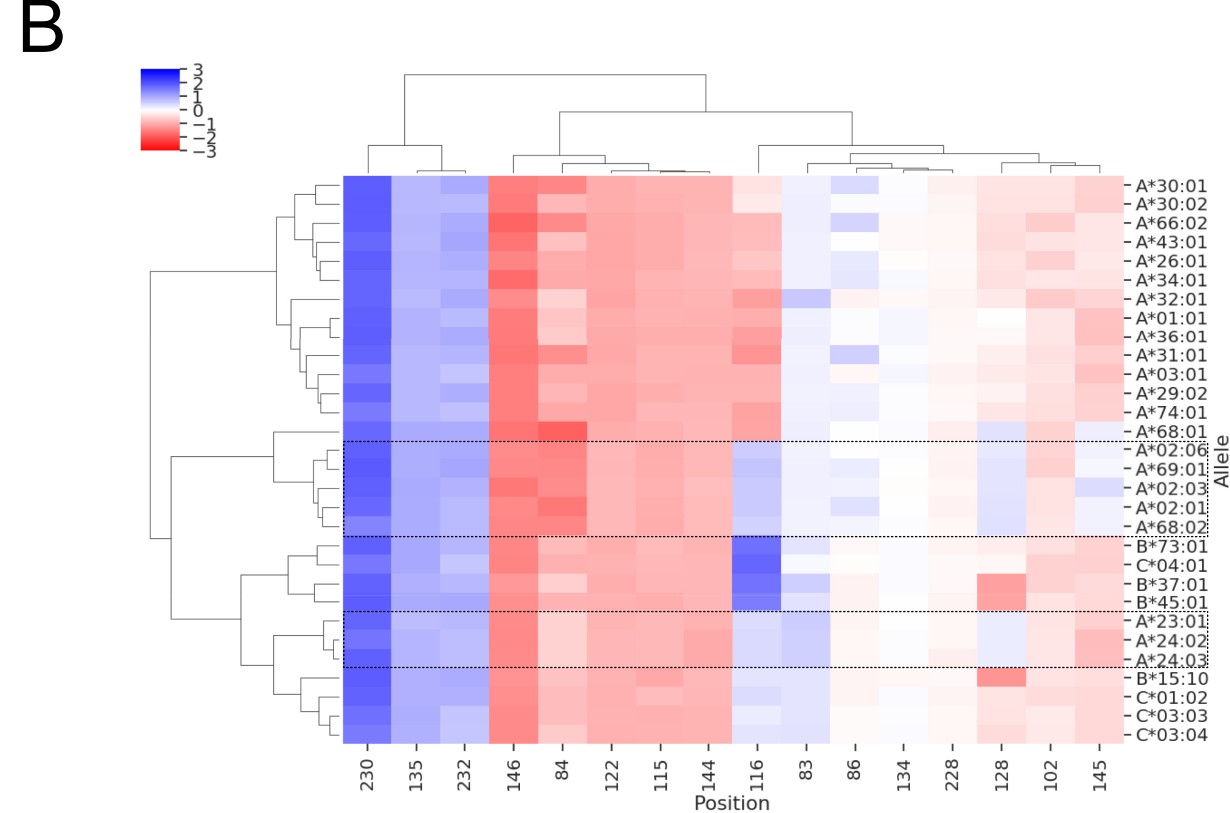

**Fig 11. Clustering of allele-specific SRFI profiles based on.** (A) peptide-binding pocket and (B) TAPBPR interface residues of 30 TAPBPR binders and non-binders as reported by Ilca et al. [101]. Position 116 was also included in clustering shown in B. TAPBPR interface residues were based on a previously reported structure of a TAPBPR-MHC I complex. Top TAPBPR binders are highlighted with dashed-line rectangles.

Here, *I* denotes information content, $p_a$ is the probability of observing amino acid *a* at the respective sequence position (calculated from input multiple sequence alignment) and $q_a$ is the pre-defined (background) probability of observing amino acid at the given position. An equal probability was used for each amino acid type (1/20).

The Evolutionary Trace (ET) method is an improvement over the classical IC approach described above [56,57]. In this method, the IE is calculated after constructing a phylogenetic tree of sequences and for each branch (node) of the tree. Then, the conservation/variation level at a given position is calculated using the following equation:

$$\rho_i = 1 + \sum_{n=1}^{N-1} \frac{1}{n} \sum_{g=1}^{n} \{ - \sum_a p_a \cdot log_2 p_a / q_a \}$$

Here, *N* is the number of sequences in the alignment and *N*−1 is the number of possible nodes in the phylogenetic tree. The final score $\rho_i$ is also termed "real-value Evolutionary Trace" score (rvET), and denotes the rank of a position (i) with respect to all other positions. Hence, lower rvET values indicate a higher evolutionary importance and vice-versa.

## Prediction of peptide binders to class I HLA alleles and homology modeling of pMHC structures

Due to the limited number of HLA alleles with structure information at the Protein Data Bank (PDB), homology modeling was used to generate pMHC structures for 1436 alleles selected as follows. For selecting peptide ligands for homology modeling, a data-driven peptide-binding prediction tool (netMHCpan 3.0) was used [64] (stand-alone version of netMHCpan 3.0 was downloaded from http://www.cbs.dtu.dk/cgi-bin/nph-sw_request?netMHCpan). First, 25000 random nonamer peptide sequences were generated using equal probability for each of the twenty naturally occurring amino acids at each peptide position. Then, binding affinities of all generated peptide sequences were predicted for each HLA allele recognized by netMHCpan 3.0 and with identified α-3 domain sequence. The results were then filtered to extract up to 10 peptides with the highest binding affinities (i.e. lowest binding free energies) and classified as strong binders by netMHCpan 3.0 for each allele. These peptides were then homology modelled in the context of HLA alleles as well as β2m using Modeller version 9.19 [106]. An X-ray structure of the HLA-B*53:01 allele was used as template (PDB ID: 1A1M).

## Local frustration analysis

Local frustration analysis was performed on all produced homology models using *frustratometer2* tool [67] (Stand-alone version available from https://github.com/gonzaparra/frustratometer2 was used). This tool uses the AWSEM (Associative Memory, Water Mediated, Structure and Energy Model) coarse-grained potential [107] to calculate residue-residue interaction energies. A sequence separation of 3 was used to calculate local amino acid densities, as defined by AWSEM. In addition to the interactions predicted by the AWSEM, long-range electrostatic interactions were also included—as offered by *frustratometer2*—using a Debye-Hückel potential:

$$V_{DH} = K_{elect} \sum_{i<j} \frac{q_i q_j}{\epsilon_r} e^{-r_{ij}/l_D}$$

where $q_i$ and $q_j$ are charges of residues $i$ and $j$, $r_{ij}$ is the distance between residues $i$ and $j$, $\epsilon_r$ is the dielectric constant of the medium (in vacuum this value is 1) and $l_D$ is the Debye-Hückel screening length, which is calculated using physiological values of temperature (25°C), dielectric constant of water (80) and ionic strength of water (0.1 M), yielding a $l_D$ of 10 Å. Here, it is possible to define an electronic strength of the protein system using a parameter called electrostatic constant ($k$):

$$k = {}^{K_{elect}}\!/_{\epsilon_r}$$

Here, a $k$ value of 4.15, which corresponds to an aqueous solution, was used.

Once the energies are computed using the AWSEM potential, *frustratometer2* assesses the local frustration present in each residue-residue contact. Here, the contacts between two residues are compared to contacts in decoy structures or molten globule configurations. The tool can produce decoys by simultaneously mutating both residues to all other amino acids. A normalization is applied in order to compare the energies of the native structure to decoys. A "mutational frustration index" ($F_{ij}^m$) then captures how favorable the contacts of the residues present in the native structure to decoys as follows:

$$F_{ij}^m = \frac{E_{i,j}^{T,N} - \langle E_{i',j'}^{T,U} \rangle}{\sqrt{1/N \sum_{k=1}^{n} \left( E_{i',j'}^{T,U} - \langle e_{i',j'}^{T,U} \rangle \right)^2}}$$

where $E_{i,j}^{T,N}$ is the total energy of native protein and $E_{i',j'}^{T,U}$ is the average energy of decoy structures. By changing the amino acid identity at both positions simultaneously, not only the contact between the respective two residues are changed, but those contacts made by the two residues with other residues are changed as well.

This index can also be calculated by changing the amino acid type of *only one of the residues* instead of applying simultaneous mutations at both positions. The index then represents how favorable the contacts made by a given amino acid are at a given position in the structure. When applied this way, the index is termed "Single Residue Frustration Index" (SRFI).

## Frustration-based clustering of HLA class I alleles

Upon application of local frustration analysis on pMHC structures, SRFI profiles averaged over peptides per HLA allele are obtained. This is represented in the form of an "SRFI matrix" with rows corresponding to positions (residues) in pMHC structure and columns corresponding to HLA alleles. A column-wise agglomerative clustering was applied on this matrix to identify similarities and differences between different HLA alleles in terms of their local structural energetics. Distances between clusters were defined by the "single linkage" criteria, in which intercluster distances were taken as the shortest distance between any two points of a pair of clusters:

$$L(r, s) = \min(D(x_{ri}, x_{sj}))$$

where $L(r,s)$ is the inter-cluster distance between clusters $r$ and $s$ and $D(x_{ri}, x_{sj})$ is the distance between points $x_{ri}$ and $x_{sj}$ of the two clusters. Here, each data point represents the SRFI profile of each HLA allele. The distance between data points were computed using Manhattan distance:

$$D(x_{ri}, x_{sj}) = \sum_{k}^{n} |x_{ri,k} - x_{sj,k}|$$

Here, the summation is performed over rows of data (k → n), which are actually positions in the pMHC structure.

## Supporting information

**S1 Table. HLA I alleles for which homology models were generated.** Clusters are also indicated with numbers (1, 2, or 3). HLA core alleles (taken from Robinson et al. [54]) are shown in bold.
(DOCX)

**S2 Table. Median SRFI and rvET values of selected positions in HLA I peptide binding groove and α-3 domain as shown in Fig 5 and Fig 7.** The rows are ordered according to increasing rvET.
(DOCX)

## Acknowledgments

The numerical calculations reported in this paper were partially performed at TUBITAK ULAKBIM, High Performance and Grid Computing Center (TRUBA resources). OS would like to acknowledge access to computing resources at RTEU Samsung Center of Excellence (SMM) for data analysis.

## Author Contributions

**Conceptualization:** Onur Serçinoğlu.

**Data curation:** Onur Serçinoğlu.

**Formal analysis:** Onur Serçinoğlu.

**Funding acquisition:** Pemra Ozbek.

**Investigation:** Onur Serçinoğlu.

**Methodology:** Onur Serçinoğlu.

**Project administration:** Pemra Ozbek.

**Resources:** Onur Serçinoğlu, Pemra Ozbek.

**Software:** Onur Serçinoğlu.

**Supervision:** Pemra Ozbek.

**Writing – original draft:** Onur Serçinoğlu.

**Writing – review & editing:** Onur Serçinoğlu.

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
