## [Decision Letter · Decision Letter 0]

4 Feb 2020

PONE-D-19-35019

Sequence-structure-function relationships in class I MHC: a local frustration perspective

PLOS ONE

Dear Dr Ozbek

Thank you for submitting your manuscript to PLOS ONE. After careful consideration, we feel that it has merit but does not fully meet PLOS ONE’s publication criteria as it currently stands. Therefore, we invite you to submit a revised version of the manuscript that addresses the points raised during the review process.

All reviewers were of the opinion that the manuscript was of interest to the field of MHC biology but have raised points that need to be addressed. Please address all the point raised with especial consideration to those raised by reviewer 3. 

We would appreciate receiving your revised manuscript by Mar 20 2020 11:59PM. To enhance the reproducibility of your results, we recommend that if applicable you deposit your laboratory protocols in protocols.io, where a protocol can be assigned its own identifier (DOI) such that it can be cited independently in the future. For instructions see: http://journals.plos.org/plosone/s/submission-guidelines#loc-laboratory-protocols

We look forward to receiving your revised manuscript.

Kind regards,

Antony Nicodemus Antoniou, PhD

Academic Editor

PLOS ONE

Journal Requirements:

1. We note that you have stated that you will provide repository information for your data at acceptance. Should your manuscript be accepted for publication, we will hold it until you provide the relevant accession numbers or DOIs necessary to access your data. If you wish to make changes to your Data Availability statement, please describe these changes in your cover letter and we will update your Data Availability statement to reflect the information you provide.

Reviewers' comments:

Reviewer's Responses to Questions

**Comments to the Author**

1. Is the manuscript technically sound, and do the data support the conclusions?

Reviewer #1: Yes

Reviewer #2: Yes

Reviewer #3: Partly

Reviewer #4: Yes

2. Has the statistical analysis been performed appropriately and rigorously? 

Reviewer #1: N/A

Reviewer #2: Yes

Reviewer #3: I Don't Know

Reviewer #4: I Don't Know

3. Have the authors made all data underlying the findings in their manuscript fully available?

Reviewer #1: Yes

Reviewer #2: Yes

Reviewer #3: Yes

Reviewer #4: Yes

4. Is the manuscript presented in an intelligible fashion and written in standard English?

Reviewer #1: Yes

Reviewer #2: Yes

Reviewer #3: Yes

Reviewer #4: Yes

5. Review Comments to the Author

Reviewer #1: The authors investigated the frustration in sequences and models of HLA-A, B and C alleles to understand the relationship between sequence, structure and function. They found higher frustration near TCR, KIR and tapasin contact positions on alpha-1 and alpha-2 helices and suggested is as a basis for the interaction. The authors also determined frustration of the complex with and without peptide. They found that the reduction in frustration upon peptide binding was highest near the F-pocket, confirming the importance of F pocket for stable peptide binding. They also found higher frustration in the HLA-C group, compared with HLA-A and B groups, especially at binding pocket positions of 66, 74 and 80 and proposed that it may affect peptide binding by HLA-C, resulting in lower stability and reduced cell surface expression. This is a comprehensive analysis of frustration in MHC class I alleles that adds to our understanding of the function of micropolymorphisms in MHC class I molecules.

Reviewer #2: The manuscript by Sercinoglu and Ozbek have described a comprehensive study using local frustration analysis to dissect the structural-functional relationship of the highly polymorphic MHC class I molecules. The study appears technically sound and adds further details to the structural characteristics of MHC class I molecules. In addition the analysis does appear to confirm much published experimental work such as the sequences which are in high frustration areas being associated with various interaction sites such TCR, KIR and tapasin, whilst at the same time presenting data across thousands of alleles and subtypes.

There are a few issues that I feel the authors need to address to clarify their manuscript.

Figure 1; a more detailed figure legend is required which illustrates the exact molecule and where the structure was obtained from.

Figure 2; to the non-expert it would be useful to generate a table highlighting the binding groove positions and the variable, hypervariable and conserved positions, especially those listed within the text. Also a table highlighting perhaps those positions with the highest and lowest rvET scores would be useful and would extract some of the pertinent findings from the analysis. Such tables may also complement Figure 3.

Figure 3; on the version I have the numerals are not clear as with most figures.

Figure 5; could the authors highlight the HLA-B*46 group of alleles within the figure as it is not obvious to the non-expert as to what alleles are being referred to.

Figure 6; a more extensive figure legend and explanation of the data within the text is recommended.

There were a few minor typographical and grammatical errors which are highlighted below;

Tapasin - small t

Line 154- 'tree' not 'three'

Line 170- sentence needs changing

Line 210-a brief definition of what the authors mean by 'energically active'

Line 211-212- sentence needs changing

Line 303- remove an ‘also’

Reviewer #3: Review of the manuscript “ Sequence-structure-function relationships in class I MHC: a local frustration perspective” by Onur Serçinoğlu and Pemra Ozbek

In this manuscript, authors Sercinoglu and Ozbek are reporting on a comparative analysis of various HLA Class I alleles, based on amino acid sequence variation, calculated local energetic frustration and evolutionary importance per each position within alpha 1 and alpha 2 domains. They claim that lesser variation at certain positions correlates with low local frustration. Such residues are likely to contribute to the formation of a conserved “MHC Class I fold” structure and consequently carry high evolutionary importance. Many other residues present intermediate to high local frustration and suggested by the authors to be involved in binding to peptide ligands, intracellular chaperones as well as receptor molecules from immune cells that survey antigen presentation by MHC Class I.

I think that this study is a great attempt to compare a vast number of HLA I sequences and to extract valuable information that offers a great potential for generating testable hypotheses that will allow investigation of MHC Class I molecules by biochemical, immunological or cell biological methods. Nevertheless, I have some major concerns, which I think should be addressed by the authors before the manuscript can be published.

1. Authors acknowledge that HLA Class I molecules are the most polymorphic set of gene products and the polymorphisms are concentrated in alpha 1 and alpha 2 domains. This is reflected in the analysis of sequence variations within those domains (Figure 2). Authors mention that the positions with high sequence variation also yield high rvET scores and vice versa. The findings propose that the two analyses of sequence variation and evolutionary trace correlate. It is not clear which residues qualify as evolutionarily conserved/important. Apart from the two cysteines that are essential for folding MHC Class I molecules, which other residues are important for forming the conserved “MHC fold”? Authors should explain clearly how their data supports their reasoning of MHC Class I structure conservation through evolution and which positions are required for folding and which others are required for MHC I function/interactions with other proteins.

2. Based on the assumptions of the authors, lower level of variance and higher evolutionary conservation is expected to be seen among alpha 3 domains of HLA I. Alpha 3 domain sequence comparisons should be included to be able to demonstrate the proposed sequence conservation during evolution for residues that are involved in the core architecture of MHC Class I.

3. For the first round of analyses performed and shown in Figure 2, I suggest that the authors compare HLA-A, HLA-B and HLA-C allele sequences within three separate groups (they only apply this separation to local frustration analysis), as those allotypes are products of three different gene loci and sequence evolution might have occurred in different positions in time. This might increase the resolution of the sequence variation analysis greatly.

4. Authors mention high levels of local frustration for residues that might be contacting TCR or tapasin. While it is conceivable that parts of MHC Class I that interact with TCR are still mobile and energetically active, it is not clear why residues that are on the tapasin interface are highly frustrated. Authors model peptide-bound MHC Class I structures where, residues contacting peptides show lesser or neutral frustration profile. After acquiring a strongly bound ligand, MHC Class I molecules are not expected to interact with tapasin so it remains to be clarified why peptide-loaded MHC Class I molecules remain frustrated in alpha2-1 region.

5. One way of calculating the contribution of peptide binding to reducing local frustration would be comparing the local frustration of the residues in alpha 1 and alpha 2 domains in the absence versus the presence of peptide in the models. Authors perform this analysis and show some results in Figure 6 but it is not possible to understand the changes in SRFI values are statistically significant. Also, the choice of residues is not clearly explained, why are residues such as 114 and 116 not included in the analysis?

Minor points:

1. The abbreviation of beta 2-microglobulin should not include any dashes (β2m).

2. In line 75, the term “HLA chain of MHC” should be revised. It is not clear what is meant by this phrase.

3. In line 84, term “protein alleles” should be revised as allelic forms of genes but proteins exist.

4. Although it is conceivable that the abbreviations “HLA” or “HLA I” or “MHC” are used by the authors to indicate HLA Class I, I advise the consistent use of one single abbreviation for the proteins.

a. Clustering data (fig5) already indicates that alleles from different loci have distinct properties and frustration profiles.

5. It is not clear in Figure 4, which position the SRFI value is derived from. Why is there a decreasing trend?

6. Figure 5A – labels on top and left of the figure are missing.

Reviewer #4: This study by Sercinoglu and Ozbek takes an interesting and unique theoretical approach based on the analysis of single residue frustration analysis (SRFI), to address the structure-function question relating to the highly diverse and evolutionarily non-conserved MHC-I protein. The paper is thought-provoking and reasonably well presented, but the following points need to be addressed before it is ready for publication:

1. In order to improve the access of this article to a more biological readership, the authors need to clarify what local frustration analysis is, what it measures and what its significance is with respect to MHC diversification during evolution. This clarification should begin within the body of the abstract (expand on ll22), and continue in the introduction and discussion.

2. A significant yet unavoidable limitation of the study is the large number of homology-modelled structures. More attention should be paid to this limitation in the text.

3. Around l176: Is it likely that higher frustration in the binding pockets (especially but not exclusively the F pocket) leads to more dependence on cofactors for assembly? That is to say, is there a correlation between frustration index and tapasin or TAPBPR dependence among the different alleles? This issue should be discussed.

4. Figure 6: Similarly, when the authors undertookSRFI on “empty” MHC alleles, they found frustration to increase. Does the degree of change in SRFI correlate with tapasin or TAPBPR dependence? This issue should be discussed.

Minor points

5. Ll30: MHC refers to the genetic locus, not the “protein complex”

6. Ll31 “of genetics” not “on genetics”

7. Ll170 necessary [to] model

6. PLOS authors have the option to publish the peer review history of their article (what does this mean?). If published, this will include your full peer review and any attached files.

Reviewer #1: No

Reviewer #2: No

Reviewer #3: Yes: Zeynep Hein

Reviewer #4: Yes: Timothy J. Elliott

---

## [Author Response · Author response to Decision Letter 0]

1 Apr 2020

We are grateful for the time and energy you expanded on our behalf. In the following sections you will find our responses to each of your points and suggestions. 

RESPONSES TO REVIEWER #1

Reviewer #1: The authors investigated the frustration in sequences and models of HLA-A, B and C alleles to understand the relationship between sequence, structure and function. They found higher frustration near TCR, KIR and tapasin contact positions on alpha-1 and alpha-2 helices and suggested is as a basis for the interaction. The authors also determined frustration of the complex with and without peptide. They found that the reduction in frustration upon peptide binding was highest near the F-pocket, confirming the importance of F pocket for stable peptide binding. They also found higher frustration in the HLA-C group, compared with HLA-A and B groups, especially at binding pocket positions of 66, 74 and 80 and proposed that it may affect peptide binding by HLA-C, resulting in lower stability and reduced cell surface expression. This is a comprehensive analysis of frustration in MHC class I alleles that adds to our understanding of the function of micropolymorphisms in MHC class I molecules.

We appreciate these positive comments by the Reviewer regarding the significance of our study. We are also grateful for the time and energy expanded on our behalf. 

RESPONSES TO REVIEWER #2

Reviewer #2: The manuscript by Sercinoglu and Ozbek have described a comprehensive study using local frustration analysis to dissect the structural-functional relationship of the highly polymorphic MHC class I molecules. The study appears technically sound and adds further details to the structural characteristics of MHC class I molecules. In addition the analysis does appear to confirm much published experimental work such as the sequences which are in high frustration areas being associated with various interaction sites such TCR, KIR and tapasin, whilst at the same time presenting data across thousands of alleles and subtypes.

There are a few issues that I feel the authors need to address to clarify their manuscript.

Figure 1; a more detailed figure legend is required which illustrates the exact molecule and where the structure was obtained from.

We’d like to apologize for any confusion this may have caused. Necessary information is now provided in the legend. 

Figure 2; to the non-expert it would be useful to generate a table highlighting the binding groove positions and the variable, hypervariable and conserved positions, especially those listed within the text. Also a table highlighting perhaps those positions with the highest and lowest rvET scores would be useful and would extract some of the pertinent findings from the analysis. Such tables may also complement Figure 3.

We’d like to thank Reviewer for this suggestion. We have constructed a Supplementary Table (Suppl. Table 2), and referred to this Table for convenience in the revised manuscript text. This table includes rvET and SRFI (median) values for binding groove as well as the alpha-3 domain residues for which data is shown in Fig 5 and Fig 7 in the revised manuscript.

Figure 3; on the version I have the numerals are not clear as with most figures.

We’d like to apologize if this caused any invonvenience. The resolution of all the Figures were increased to overcome this issue. 

Figure 5; could the authors highlight the HLA-B*46 group of alleles within the figure as it is not obvious to the non-expert as to what alleles are being referred to.

Figure 6; a more extensive figure legend and explanation of the data within the text is recommended.

We’d like to thank Reviewer for highlighting this issue. We’ve used the python package seaborn for constructing the clustermaps. The allele names on this clustermap figure (Fig 9 now) do not contain all alleles we have studied for visualization reasons. This is automatically taken care of by the seaborn package, and without really modifying the source code of this package, we are unable to highlight the row of a specific allele or allele groups.

Those HLA-B*46 group alleles that are clustered along with HLA-C alleles can be seen in Supplementary Table 1. 

There were a few minor typographical and grammatical errors which are highlighted below;

Tapasin - small t

Line 154- 'tree' not 'three'

Line 170- sentence needs changing

We’d like to thank Reviewer for capturing these mistakes. They have been corrected, and necessary changes were made to the sentence on Line 170 (which Is on line 200 in the revised manuscript).

Line 210-a brief definition of what the authors mean by 'energically active'

We have chosen to remove this statement to avoid confusion, as even a brief explanation would involve the description of the computational method in question, and would divert the attention of the reader. 

Line 211-212- sentence needs changing

Line 303- remove an ‘also’

Necessary changes were made. 

 

RESPONSES TO REVIEWER #3

In this manuscript, authors Sercinoglu and Ozbek are reporting on a comparative analysis of various HLA Class I alleles, based on amino acid sequence variation, calculated local energetic frustration and evolutionary importance per each position within alpha 1 and alpha 2 domains. They claim that lesser variation at certain positions correlates with low local frustration. Such residues are likely to contribute to the formation of a conserved “MHC Class I fold” structure and consequently carry high evolutionary importance. Many other residues present intermediate to high local frustration and suggested by the authors to be involved in binding to peptide ligands, intracellular chaperones as well as receptor molecules from immune cells that survey antigen presentation by MHC Class I.

I think that this study is a great attempt to compare a vast number of HLA I sequences and to extract valuable information that offers a great potential for generating testable hypotheses that will allow investigation of MHC Class I molecules by biochemical, immunological or cell biological methods. Nevertheless, I have some major concerns, which I think should be addressed by the authors before the manuscript can be published.

1. Authors acknowledge that HLA Class I molecules are the most polymorphic set of gene products and the polymorphisms are concentrated in alpha 1 and alpha 2 domains. This is reflected in the analysis of sequence variations within those domains (Figure 2). Authors mention that the positions with high sequence variation also yield high rvET scores and vice versa. The findings propose that the two analyses of sequence variation and evolutionary trace correlate. It is not clear which residues qualify as evolutionarily conserved/important. Apart from the two cysteines that are essential for folding MHC Class I molecules, which other residues are important for forming the conserved “MHC fold”? Authors should explain clearly how their data supports their reasoning of MHC Class I structure conservation through evolution and which positions are required for folding and which others are required for MHC I function/interactions with other proteins.

We’d like to thank the Reviewer for this question. 

We have indeed used rvET as a measure of sequence conservation (with lower rvET values indicating higher conservation) as well as evolutionary importance. We’d like to refer to our response to your third comment below for an explanation of how the rvET identifies evolutionarily important positions.

On the other hand, a residue at a specific position may be conserved for different reasons, e.g. for maintaining stability, catalysis, protein/ligand interactions, etc. We used the local frustration analysis to quantify how energetically favorable interactions of the respective residue are in the context of the given structure. Here, minimal frustration (i.e. high SRFI) usually indicates a residue that is important for protein stability. Finally, we identified the residues/positions that are essential for folding MHC based on their SRFI values and rvET scores. Accordingly, minimally-frustrated and conserved residues should be functionally important for formation of the conserved MHC formation and stability. In contrast, highly frustrated residues (i.e. low SRFI) may be involved in protein-protein interactions. 

We have now revised the text under “Integrating biophysics into HLA I evolution” heading of Results section to include further explanatory statements regarding how SRFI profiles and rvET (sequence conservation) values were interpreted to reflect the functional importance of HLA residues. For clarity, we also included an additional Figure (Fig 8) to better reflect how minimally-frustrated and conserved residues are physically connected within the HLA structure.

2. Based on the assumptions of the authors, lower level of variance and higher evolutionary conservation is expected to be seen among alpha 3 domains of HLA I. Alpha 3 domain sequence comparisons should be included to be able to demonstrate the proposed sequence conservation during evolution for residues that are involved in the core architecture of MHC Class I.

We’d like to thank the Reviewer for highlighting this issue. 

We largely focused on the HLA peptide-binding groove, as this is the region where the highest level of polymorphism level is observed. Moreover, the rvET values are highly dependent on the input sequence alignments, and including as many different allele sequences in this analysis as possible is important to obtain more reliable scores. Since there is a large gap between the number of alleles with and without alpha-3 domain sequences (8696 versus 1436), we had included only binding-groove sequences in our initial analysis. 

Based on your suggestion, we performed the same sequence variation and real-value Evolutionary Trace analysis for the alpha 3 domain as well. The results are given in Figure 3, and necessary additions were made in the manuscript text. Note that the sequence variation analysis involved only the 1436 HLA alleles for which homology models could be generated (as opposed to the 8696 HLA allele sequences for which the binding groove sequence is available), since only this many alleles featured the sequence of their alpha-3 domains. 

3. For the first round of analyses performed and shown in Figure 2, I suggest that the authors compare HLA-A, HLA-B and HLA-C allele sequences within three separate groups (they only apply this separation to local frustration analysis), as those allotypes are products of three different gene loci and sequence evolution might have occurred in different positions in time. This might increase the resolution of the sequence variation analysis greatly.

We’d like to thank the Reviewer for pointing out this aspect of the sequence variation analysis. 

Your suggestion is quite valid: the evolution of HLA allele sequences should indeed be considered in the sequence variation analysis. The sequence variation analysis in our study may appear not to take this into account, yet the metric we have chosen to identify conserved and "evolutionarily important" residues/positions within the HLA sequence, the "real-value Evolutionary Trace" (rvET) is actually based on both variation level of each position in sequence and where exactly the variation occurs along a phylogenetic tree that is constructed from input sequences. As such, the rvET should be particulary suitable for studying sequence variation in HLA genes. 

Here is an example from the reference paper of the evolutionary trace method (Fig 1 from Wilkins et al., 2012). Suppose we have the following sequence alignment, along with the associated phylogenetic tree:

Figure 1 from Wilkins et al. (2012), Figure taken from PubMed Central (https://www.ncbi.nlm.nih.gov/pmc/articles/PMC4892863/)

If we were to quantify sequence variation/conservation only by using a more “conventional” metric such as Shannon’s entropy, and attribute equal weight to each amino acid type, positions 3 and 4 (which predominantly feature Q and V amino acids) should be considered equivalent in terms of how conserved they are. The Evolutionary Trace method however will prioritize position 3 over position 4 in this case, as “the basic hypothesis behind the method is that “residues that vary among widely divergent branches of evolution are more likely to have a larger functional impact than other residues that vary even among closely related species”. Even though here the main aim is to enable a better comparison of sequences from different species, the same logic could be applied to comparisons of HLA alleles sequences that are included in different gene loci.

The main idea/purpose of our study is based on a comparison of the conserved MHC fold and HLA sequence variation. Our knowledge of the MHC protein complex structures suggests that this fold is conserved across different allele groups, hence there is a single well-defined shape of the complex. We also aimed to identify positions that are functionally important (i.e. to maintain the shape/stability as well as protein interactions) within this fold. Therefore, we used a single metric to quantify how important each position is for this conserved fold. In that sense, we believe that the choice of using rvET based on an analysis of HLA binding groove sequences of all HLA-A, HLA-B and HLA-C allele groups, is justified.

On the other hand, one may argue that the amino acid differences between allele groups that contribute to the clustering pattern based on SRFI profiles remained unclear in the manuscript text. We have supplemented the Figure showing the SRFI-based clustering results (Figure 10) with sequences logos that highlight differences between amino acids included in clustering positions of different SRFI clusters (14 positions shown in the SRFI clustermap in Figure 10A). For this purpose, we used the Two Sample Logos server (twosamplelogo.org, Vacic et al. (2006)). 

Wilkins, A., Erdin, S., Lua, R., & Lichtarge, O. (2012). Evolutionary trace for prediction and redesign of protein functional sites. Methods in molecular biology (Clifton, N.J.), 819, 29–42. https://doi.org/10.1007/978-1-61779-465-0_3

Vacic V, Iakoucheva LM, Radivojac P. Two Sample Logo: a graphical representation of the differences between two sets of sequence alignments. Bioinformatics. 2006;22: 1536–1537. doi:10.1093/bioinformatics/btl151

4. Authors mention high levels of local frustration for residues that might be contacting TCR or tapasin. While it is conceivable that parts of MHC Class I that interact with TCR are still mobile and energetically active, it is not clear why residues that are on the tapasin interface are highly frustrated. Authors model peptide-bound MHC Class I structures where, residues contacting peptides show lesser or neutral frustration profile. After acquiring a strongly bound ligand, MHC Class I molecules are not expected to interact with tapasin so it remains to be clarified why peptide-loaded MHC Class I molecules remain frustrated in alpha2-1 region.

We’d like to thank the Reviewer for bringing our attention to this issue regarding peptide-binding related changes in local frustration. 

Our current knowledge indeed tells us that Tapasin functions as a chaperone to keep the peptide-free MHC in a peptide-receptive state, and that the Tapasin-MHC interaction is broken once a high-affinity peptide is loaded by the peptide-binding groove. This has also been demonstrated to involve a molecular “tug-of-war” mechanism between the peptide C-terminus and Tapasin through their interactions with the alpha-2-1 region (Fisette et al., 2016). 

How do we link high frustration in the alpha2-1 region of peptide-loaded MHC with Tapasin interaction, when logic tells us the frustration should be reduced upon peptide-binding? Here, we’d like to highlight what exactly the local frustration metric we used, the SRFI, tells us, and further clarify why it is relevant for sites involved in binding to Tapasin. We used the frustratometer2 tool in our study. Frustratometer2 computers the SRFI metric via “randomizing the identity of every single amino acid within the decoy structure set, and thus evaluates every possible mutation at every site in a well-defined native structure” (R.Parra et al. (2016)). In other words, the SRFI is a relative measure, and answers only the following question: “Among all other amino acids at a specific location within a protein structure, how favorable/unfavorable the present amino acid is terms of its contacts with surrounding residues?”.

In the context of Tapasin-MHC interaction, we may re-phrase this question as “Among all other amino acids that may be present in the alpha2-1 region, how favorable are the given amino acids in our homology models in terms of contacting Tapasin molecule?”. Considering the related literature information on patches of residues with increased frustration on protein surfaces involved in binding (Freiberger et al., 2019; Ferreiro et al., 2014), we may argue that our observation of increased frustration at Tapasin contacts sites in our homology models indicates that human MHC, the HLA, evolves to maintain contacts with Tapasin. 

Hence, the frustration analysis we present only indicates that the amino acids in the alpha2-1 region and other sites contacting Tapasin are “optimal” for protein interaction. However, even though it is an important prerequisite, the residue identity is not enough to explain protein-protein interaction, as these interactions are usually transient (as in the case of TCR-pMHC interactions). An investigation of whether HLA and Tapasin will remain in contact upon peptide-binding should consider conformational changes and protein dynamics into account. Furthermore, other appropriate measures that are not relative measures such as SRFI should be used for this purpose. For example, configurational frustration may be used to describe local frustration after simulating peptide-free and -bound MHC structures and obtaining relevant conformations in such a study. 

Parra RG, Schafer NP, Radusky LG, Tsai MY, Guzovsky AB, Wolynes PG, et al. Protein Frustratometer 2: a tool to localize energetic frustration in protein molecules, now with electrostatics. Nucleic Acids Res. 2016;44: W356–W360. doi:10.1093/nar/gkw304

Fisette O, Wingbermühle S, Tampé R, Schäfer L V. Molecular mechanism of peptide editing in the tapasin-MHC I complex. Sci Rep. 2016;6: 19085. doi:10.1038/srep19085

Freiberger MI, Brenda Guzovsky A, Wolynes PG, Gonzalo Parra R, Ferreiro DU. Local frustration around enzyme active sites. Proc Natl Acad Sci U S A. 2019;116: 4037–4043. doi:10.1073/pnas.1819859116

Ferreiro DU, Komives EA, Wolynes PG. Frustration in biomolecules. Quarterly Reviews of Biophysics. Cambridge University Press; 2014. pp. 285–363. doi:10.1017/S0033583514000092

5. One way of calculating the contribution of peptide binding to reducing local frustration would be comparing the local frustration of the residues in alpha 1 and alpha 2 domains in the absence versus the presence of peptide in the models. Authors perform this analysis and show some results in Figure 6 but it is not possible to understand the changes in SRFI values are statistically significant. Also, the choice of residues is not clearly explained, why are residues such as 114 and 116 not included in the analysis?

We thank the Reviewer for highlighting this issue regarding the statistical significance of SRFI change upon peptide-binding. 

The residues shown in this Figure (in the current version, Fig 10) are the top 10 residues that show the highest increase in SRFI upon peptide binding (please see the following to see how exactly the SRFI change was computed). In fact, all residues, including those at positions 114 and 116 were included in the analysis, they were just not among those in top 10 (position 116 ranked 220th, and 114 205th, respectively). 

A direct approach to compute SRFI changes upon peptide-binding here may involve simply computing differences between SRFI values of each position in peptide-bound and -free structures, ignoring allelic polymorphism. This approach, however, may be misleading, as here it is assumed the frustration profiles within the structures of each group, peptide-free and -bound states, are equivalent to each other, and hence a statistical significance test can be applied. In fact, they are not, since SRFI is a relative measure (as we describe above, and highly dependent on the allele sequence).

Therefore, we took an alternative approach and computed SRFI differences by subtracting the median SRFI in peptide-bound HLA models (up to 10 structures) and a single SRFI value in a single peptide-free HLA structure for each allele. The distributions in Figure 10 thus indicate distribution of SRFI change values (a single value from each allele for each position). 

Nevertheless, it could still be possible to assess statistical significance of SRFI change for each allele. However, this would involve multiple comparisons of ten SRFI values against a single one. Considering this limitation, and the simple fact removing the peptide from pMHC models does not lead to representative structure of the peptide-free state of MHC, we choose to refrain from performing a statistical analysis. Our findings in this section are therefore highly exploratory and should be validated or revised in more extensive and proper studies. 

We have included explanatory statements to clarify these aspects of our analysis in the manuscript. 

Minor points:

1. The abbreviation of beta 2-microglobulin should not include any dashes (β2m).

Necessary changes were made in the manuscript text.

2. In line 75, the term “HLA chain of MHC” should be revised. It is not clear what is meant by this phrase.

We’d like to apologize for any confusion this might have caused. We have made the necessary change.

3. In line 84, term “protein alleles” should be revised as allelic forms of genes but proteins exist.

We’d like to apologize for any confusion this might have caused. We have removed “protein” from this term to avoid confusion. 

4. Although it is conceivable that the abbreviations “HLA” or “HLA I” or “MHC” are used by the authors to indicate HLA Class I, I advise the consistent use of one single abbreviation for the proteins.

We’d like to apologize for any confusion this might have caused. We have made relevant changes for consistent use of abbreviations throughout the text. 

a. Clustering data (fig5) already indicates that alleles from different loci have distinct properties and frustration profiles.

We’d like to thank the Reviewer for this feedback. Do you mean that it may not be necessary to map SRFI values on binding groove structures? 

5. It is not clear in Figure 4, which position the SRFI value is derived from. Why is there a decreasing trend?

We’d like to thank the Reviewer for this question. In this Figure, (Fig 5 in the update manuscript), we delibaretly ordered positions according to decreasing SRFI values. The main idea here was to find out which positions were minimally, highly, and neutrally frustrated, and this is visually much more easily identified when the positions are ordered as such. Fig 5A highlights minimally and highly frustrated positions, whereas Fig 5B highlights the neutral frustration residues. 

6. Figure 5A – labels on top and left of the figure are missing.

We’d like to thank the Reviewer for this comment. This figure (Fig 9A in the revised manuscript) has been revised to include labels (Clusters).

 

RESPONSES TO REVIEWER #4

This study by Sercinoglu and Ozbek takes an interesting and unique theoretical approach based on the analysis of single residue frustration analysis (SRFI), to address the structure-function question relating to the highly diverse and evolutionarily non-conserved MHC-I protein. The paper is thought-provoking and reasonably well presented, but the following points need to be addressed before it is ready for publication:

1. In order to improve the access of this article to a more biological readership, the authors need to clarify what local frustration analysis is, what it measures and what its significance is with respect to MHC diversification during evolution. This clarification should begin within the body of the abstract (expand on ll22), and continue in the introduction and discussion.

We appreciate this valuable feedback.

We have provided a more extensive explanation of local frustration, and why it is particularly useful for studying HLA polymorphism in the introduction section. Our changes are highlighted in red color. Kindly refer to the changes in the revised manuscript for more detail.

2. A significant yet unavoidable limitation of the study is the large number of homology-modelled structures. More attention should be paid to this limitation in the text.

We’d like to thank the Reviewer for this comment. The section “Homology modeling of HLA alleles in the context of pMHC” have been included exactly to highlight why homology modeling was necessary in the study. We have not inserted additional two sentences describing why homology modeling was used to model peptides as well. 

3. Around l176: Is it likely that higher frustration in the binding pockets (especially but not exclusively the F pocket) leads to more dependence on cofactors for assembly? That is to say, is there a correlation between frustration index and tapasin or TAPBPR dependence among the different alleles? This issue should be discussed.

We’d like to thank the Reviewer for bringing our attention to dependence of HLA alleles on cofactors for peptide loading.

We used a recent study by Ilca et al. (Ilca et al. (2019)) as reference to investigate this issue further with respect to our frustration data. In this study, Ilca et al. compared a number of HLA alleles in terms of their TAPBPR binding levels, and found out that the TAPBPR prefers binding to HLA-A allotypes particularly in the A2 and A24 supertypes rather than HLA-B and HLA-C allotypes. 

We have inspected local frustration profiles of 30 alleles for which TAPBPR binding levels were reported by these authors based on either peptide-binding pocket residues or TAPBPR binding interface residues. We have summarized our findings under a new heading “Local frustration in the context of TAPBPR-HLA interaction”. Kindly refer to the respective section of the update manuscript for further details. 

4. Figure 6: Similarly, when the authors undertookSRFI on “empty” MHC alleles, they found frustration to increase. Does the degree of change in SRFI correlate with tapasin or TAPBPR dependence? This issue should be discussed.

We’d like to thank the Reviewer for this question.

A significant limitation here is the simple fact that peptide-free models in our study are certainly not representative of the actual structural ensembles, as it is known that in its peptide-free state, the complex does not have a well-defined three dimensional structure. Our peptide-free models were generated by simply “deleting” the peptide from the binding groove. A comparison of peptide-bound and -free models in terms of SRFI profiles here may give clues regarding which residues are immediately affected from peptide removal. However, any further investigation based on peptide-free MHC regarding the behavior of the whole complex, such as the dependency on chaperone for assembly, should necessarily involve conformation changes related to peptide removal to be modelled. Therefore, while it is perfectly reasonable and even necessary from a biological perspective, we refrained from performing this analysis regarding the SRFI change between peptide-free and -bound homology models for the above reasons.

Minor points

5. Ll30: MHC refers to the genetic locus, not the “protein complex”

We’d like to apologize for confusion here. We have revised the relevant section of the abstract as: 

“The protein complex encoded by Major Histocompatibility Complex (MHC) genes in our body cells plays a critical role in our fight against pathogens via presentation of antigenic peptides to receptor molecules of our immune system cells”

6. Ll31 “of genetics” not “on genetics”

7. Ll170 necessary [to] model

We’d like to thank the Reviewer for highlighting these mistakes. These have been corrected in the revised manuscript.

---

## [Decision Letter · Decision Letter 1]

23 Apr 2020

Sequence-structure-function relationships in class I MHC: a local frustration perspective

PONE-D-19-35019R1

Dear Dr. Ozbek,

We are pleased to inform you that your manuscript has been judged scientifically suitable for publication and will be formally accepted for publication once it complies with all outstanding technical requirements.

With kind regards,

Antony Nicodemus Antoniou, PhD

Academic Editor

PLOS ONE

Additional Editor Comments (optional):

Reviewers' comments:

Reviewer's Responses to Questions

**Comments to the Author**

1. If the authors have adequately addressed your comments raised in a previous round of review and you feel that this manuscript is now acceptable for publication, you may indicate that here to bypass the “Comments to the Author” section, enter your conflict of interest statement in the “Confidential to Editor” section, and submit your "Accept" recommendation.

Reviewer #2: All comments have been addressed

Reviewer #3: All comments have been addressed

2. Is the manuscript technically sound, and do the data support the conclusions?

Reviewer #2: Yes

Reviewer #3: Yes

3. Has the statistical analysis been performed appropriately and rigorously? 

Reviewer #2: I Don't Know

Reviewer #3: I Don't Know

4. Have the authors made all data underlying the findings in their manuscript fully available?

Reviewer #2: Yes

Reviewer #3: Yes

5. Is the manuscript presented in an intelligible fashion and written in standard English?

Reviewer #2: Yes

Reviewer #3: Yes

6. Review Comments to the Author

Reviewer #2: The authors have addressed all issues that I have raised. There are some points, mainly typographical issues that need addressing and only minor changes to figure legends.

L59; protein surfaces

L81; the cell surface

L136; What do the TEM abbreviations mean

L303; Do the authors mean Fig 5?

L314; The analysis of the a3 domain has highlighted that p203 and 259 are conserved reflecting that these positions form a conserved disulphide bond. The authors could perhaps briefly comment as they did previously for C101-C164 pairing.

Figure Legends

Fig 4; the different views of the molecular structure should be mentioned within the figure legend.

Fig 8; the legend just needs to be a bit more specific i.e. 'highlighting interactions with HLA I heavy chain"

There is a formatting issue with supplementary table 1.

Reviewer #3: I believe that the authors understood and have made considerable effort to adress the questions that I have raised during the review process. I find that the additional figures, sections and more detailed explanation of the method have contributed to the clarity and understability of the manuscript and the novel approach in the field of MHC class I bioinformatics.

7. PLOS authors have the option to publish the peer review history of their article (what does this mean?). If published, this will include your full peer review and any attached files.

Reviewer #2: No

Reviewer #3: Yes: Zeynep Hein

---

## [Editor Report · Acceptance letter]

5 May 2020

PONE-D-19-35019R1 

Sequence-structure-function relationships in class I MHC: a local frustration perspective 

Dear Dr. Ozbek:

I am pleased to inform you that your manuscript has been deemed suitable for publication in PLOS ONE. Congratulations! Your manuscript is now with our production department. 

With kind regards,

on behalf of

Dr. Antony Nicodemus Antoniou 

Academic Editor

PLOS ONE